# WorldCache: Accelerating World Models for Free via Heterogeneous Token Caching

Weilun Feng [* 1 2]  Guoxin Fan [* 1 2]  Haotong Qin [* 3]  Mingqiang Wu [1 2]  Yuqi Li [4]  Xiangqi Li [1 2]  Zhulin An [✉ 1]
Libo Huang [1]  Dingrui Wang [5]  Longlong Liao [6]  Michele Magno [3]  Yongjun Xu [1 7]  Chuanguang Yang [✉ 1]

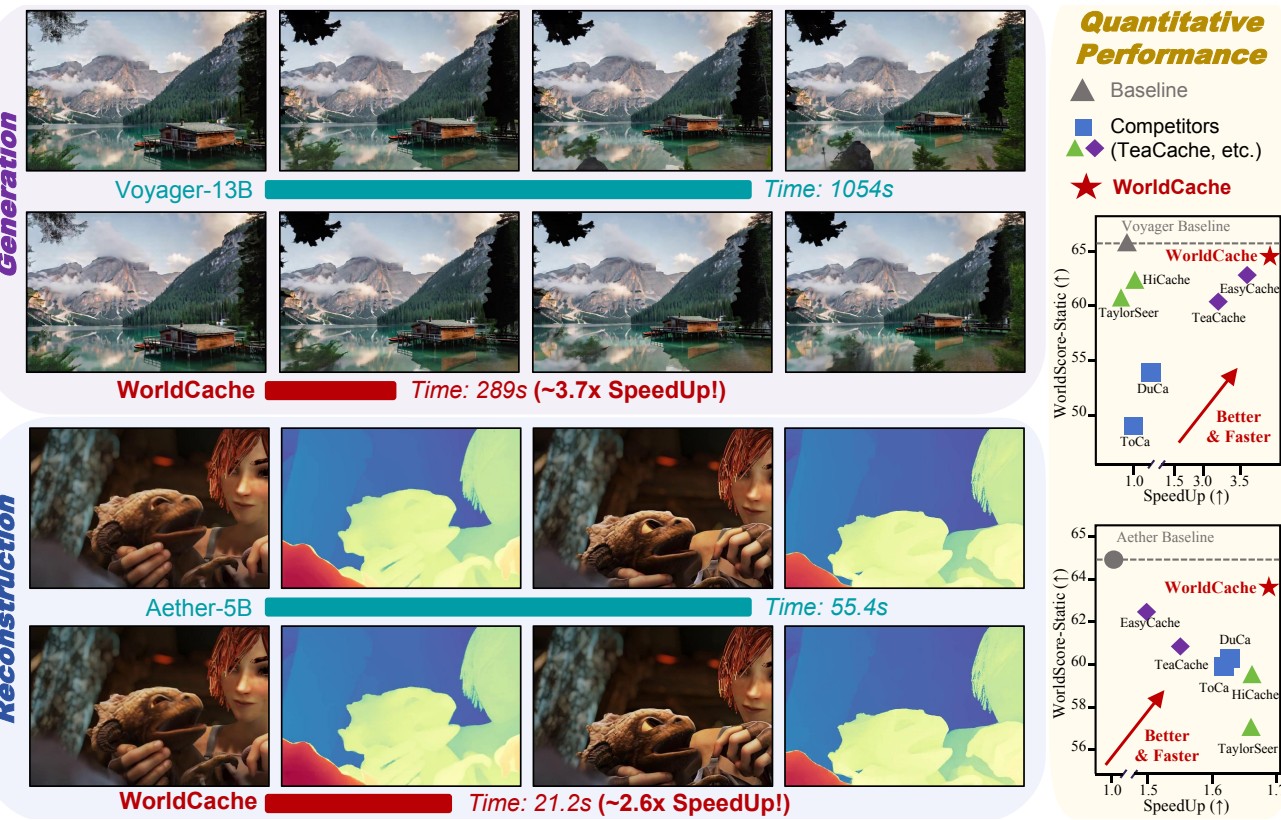

*Figure 1.* **WorldCache** greatly accelerates two diffusion world models: HunyuanVoyager (Huang et al., 2025) and Aether (Zhu et al., 2025) with up to **3.7×** speedup, while preserving high-fidelity details.

## Abstract

Diffusion-based world models have shown strong potential for unified world simulation, but the iterative denoising remains too costly for interactive use and long-horizon rollouts. While feature caching can accelerate inference without training, we find that policies designed for single-modal diffusion transfer poorly to world models due to two world-model-specific obstacles: *token heterogeneity* from multi-modal coupling and spatial variation, and *non-uniform temporal dynamics* where a small set of hard tokens drives error growth, making uniform skipping either unstable or overly conservative. We propose **WorldCache**, a caching framework tailored to diffusion world models. We introduce *Curvature-guided Heterogeneous Token Prediction*, which uses a physics-grounded curvature score to estimate token predictability and applies a Hermite-guided damped predictor for chaotic tokens with abrupt direction changes. We also design *Chaotic-*

[*]Equal contribution  [1]State Key Laboratory of AI Safety, Institute of Computing Technology, Chinese Academy of Sciences [2]University of Chinese Academy of Sciences [3]ETH Zürich [4]City College of New York, City University of New York, USA [5]Technical University of Munich [6]Fuzhou University [7]Xiamen Institute of Data Intelligence, Xiamen, China. Correspondence to: [✉]Chuanguang Yang <yangchuanguang@ict.ac.cn>, [✉]Zhulin An <anzhulin@ict.ac.cn>.

*Proceedings of the 43rd International Conference on Machine Learning*, Seoul, South Korea. PMLR 306, 2026. Copyright 2026 by the author(s).

*prioritized Adaptive Skipping*, which accumulates a curvature-normalized, dimensionless drift signal and recomputes only when bottleneck tokens begin to drift. Experiments on diffusion world models show that WorldCache delivers up to **3.7×** end-to-end speedups while maintaining **98%** rollout quality, demonstrating the vast advantages and practicality of WorldCache in resource-constrained scenarios. Our code is released in https://github.com/FofGofx/WorldCache.

## 1. Introduction

World models (Bar et al., 2025; Liu et al., 2024; Bruce et al., 2024; Agarwal et al., 2025; Hafner et al., 2025; An et al., 2025) have recently emerged as a compelling foundation for building more general-purpose intelligence. Rather than merely generating observed contents, world models aim to capture spatiotemporal dynamics of the environment, enabling long-horizon imagination for planning, decision making, and interactive agents. With the rapid progress of large-scale generative models (Croitoru et al., 2023; Naveed et al., 2025; Feng et al., 2026b), generation-driven world models (Russell et al., 2025; Bar et al., 2025; Huang et al., 2025; Zhu et al., 2025; Li et al., 2026) built upon diffusion models have gained increasing attention for synthesizing immersive, coherent, and even interactive virtual environments from large-scale data.

However, modern diffusion-based world models remain costly at inference time since they require many denoising steps with repeated backbone evaluations (Ho et al., 2020; Lipman et al., 2022). Recently, various techniques (Feng et al., 2025c; Xi et al., 2025; Feng et al., 2025b; Zhang et al., 2024; Feng et al., 2025a; 2026c; 2025d) have been developed to enable efficient diffusion inference. Among which, *feature caching* (Selvaraju et al., 2024; Ma et al., 2024b; Liu et al., 2025a) is particularly attractive due to its training-free nature: it reduces sampling cost by reusing or cheaply forecasting intermediate representations across timesteps.

While feature caching has achieved strong speedups in single-modal image or video diffusion, we identify that directly transferring existing policies to diffusion world models often leads to rapid error accumulation and unstable rollouts. In particular, world-model simulation exhibits two distinctive properties that fundamentally challenge conventional caching:

❶ **Heterogeneous token evolution with a long-tailed difficulty profile.** Unlike single-modal diffusion where token dynamics are relatively uniform, world models jointly evolve tokens that correspond to different physical factors (e.g., appearance vs. geometry) and different spatial deriva-

tion. Consequently, the *predictability* of token trajectories is highly non-uniform: most tokens evolve smoothly and are easy to reuse or extrapolate, but a small fraction exhibit sharp, non-linear changes tied to physically critical structures (e.g., motion boundaries or depth discontinuities). This long-tailed difficulty makes uniform caching inherently inefficient: a global conservative rule wastes computation on the easy majority, whereas a global aggressive rule is bottlenecked by the hard minority and causes overall drift.

❷ **Non-stationary temporal regimes where a few bottleneck tokens dominate failure.** World-model denoising is also *regime-dependent*: the model may traverse long intervals where trajectories are smooth and caching is reliable, followed by short intervals where dynamics become abruptly non-linear. Importantly, caching failure is typically triggered not by average feature change, but by the same hard-to-cache token subset becoming unpredictable in these difficult regimes. As a result, fixed skipping schedules may miss critical updates, and global-threshold heuristics that treat all tokens equally either (i) react too late when bottleneck tokens drift, or (ii) over-trigger due to benign changes in easy tokens, yielding poor speed–quality trade-offs.

To address these challenges, we present **WorldCache**, a training-free acceleration framework tailored for diffusion world models through *heterogeneous token caching*. Our approach introduces *Curvature-guided Heterogeneous Token Prediction* (CHTP), which uses a physics-grounded curvature score to estimate token-wise predictability and assigns different approximation rules: 0th-order reuse for stable tokens, 1st-order extrapolation for near-linear tokens, and a curvature-aware damped predictor for chaotic tokens. To regulate when expensive backbone evaluations are necessary, we further propose *Chaotic-prioritized Adaptive Skipping* (CAS), which constructs a *dimensionless* drift indicator by combining curvature with feature deviations. This yields a unified, scale-normalized uncertainty score whose accumulation triggers `FULL` computation precisely when the bottleneck token subset begins to drift, enabling aggressive skipping without destabilizing multi-modal rollouts.

Our contributions can be summarized as:

- We identify two world-model-specific challenges that hinder existing diffusion caching methods: long-tailed token predictability induced by multi-modal heterogeneity, and non-stationary temporal regimes where bottleneck tokens dominate caching failure.

- We propose curvature-guided heterogeneous token prediction that allocates different caching rules to tokens based on trajectory nonlinearity, with a dedicated damped predictor for chaotic tokens.

- We introduce a chaotic-prioritized adaptive skipping strategy with a curvature-induced *dimensionless*

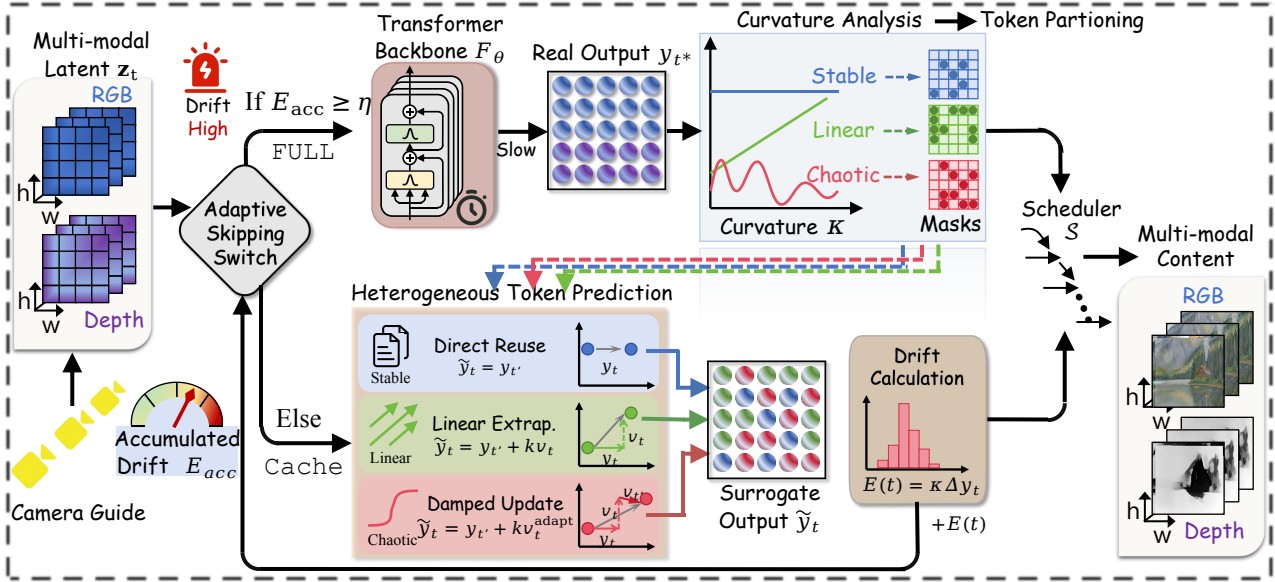

*Figure 2.* **Overview of the proposed WorldCache framework.** The pipeline alternates between `FULL` backbone evaluation and `CACHE` approximation. **(Top)** In each full computation step, tokens are partitioned into Stable, Linear, and Chaotic groups based on their curvature $\kappa$. **(Bottom)** During caching steps, heterogeneous predictors (Reuse, Linear Extrapolation, or Damped Update) are applied accordingly. **(Left)** The Chaotic-prioritized Adaptive Skipping (CAS) mechanism accumulates a curvature-normalized drift score $E_{acc}$ specifically from chaotic tokens, triggering a full computation only when critical drift is detected.

drift score, enabling a unified threshold for stable caching decisions across heterogeneous token scales and timesteps.

- Extensive experiments on diffusion world models demonstrate that WorldCache substantially reduces sampling cost while preserving multi-modal rollout quality.

## 2. Related Works

### 2.1. Data-Driven World Models

Data-driven world models (Ha & Schmidhuber, 2018; Le-Cun, 2022) learn predictive internal representations of the environment to simulate futures for control and planning. Classical methods build compact latent dynamics with recurrent state-space models, enabling imagination and policy optimization (Hafner et al., 2019; 2020; 2023; 2025). More recently, scaling laws in generative modeling have motivated *tokenized* world models that generate high-fidelity, long-horizon rollouts, including interactive environments learned from large-scale video data (Bruce et al., 2024; Agarwal et al., 2025) and large video generators discussed as emergent "world simulators" (Liu et al., 2024; Russell et al., 2025; Bar et al., 2025). Building upon generative models, diffusion-based world models further adopt DiT (Peebles & Xie, 2023) backbones to jointly model coupled modalities (e.g., RGB and geometry/depth, optionally action-conditioned) for unified world representation

and downstream simulation (Huang et al., 2025; Zhu et al., 2025). However, such unified multi-modal generation substantially amplifies inference cost, motivating acceleration techniques tailored to world-model dynamics.

### 2.2. Feature Caching for Diffusion Models

Feature caching is a training-free paradigm that accelerates diffusion sampling by exploiting temporal redundancy across denoising steps. Existing methods can be broadly grouped into: (i) *reuse-based caching* that skips computation by reusing intermediate representations across nearby steps, often at block/layer granularity (Selvaraju et al., 2024; Ma et al., 2024b;a; Kahatapitiya et al., 2025; Chen et al., 2025); (ii) *token-adaptive caching* that applies selective reuse to subset tokens while preserving other tokens for full computation (Zou et al., 2024a;b; Zheng et al., 2025); (iii) *forecasting-based caching* that predicts future features via local trajectory approximation (e.g., Taylor expansion) or trajectory integration, reducing long-interval drift (Liu et al., 2025b); and (iv) *runtime-adaptive scheduling* that decides when to cache using lightweight proxies or online uncertainty signals (Liu et al., 2025a; Zhou et al., 2025).

However, most prior caching strategies are developed for *single-modal* image/video diffusion and implicitly assume relatively homogeneous feature dynamics. This assumption becomes fragile in world models, where coupled multi-modal tokens exhibit distinct physical evolution patterns. This motivates us to introduce a token-heterogeneous

caching mechanism tailored to world models.

## 3. Preliminaries

**Diffusion World Models with Transformer Backbones.**
We consider a diffusion-based world model that generates a multi-modal world state through $T$ denoising steps, following recent transformer-based diffusion world models such as Voyager (Huang et al., 2025) and Aether (Zhu et al., 2025) (we use Voyager for illustration). Let $\mathbf{z}_t^r \in \mathbb{R}^{c \times f \times h \times w}$ denote the RGB latents for 2D video generation and $\mathbf{z}_t^d \in \mathbb{R}^{c \times f \times h \times w}$ denote the corresponding depth latents for 3D estimation at timestep $t$. $f, h, w$ denote the frame, height, and width, respectively. We form the multi-modal latent by spatial concatenation

$$\mathbf{z}_t = \mathrm{concat}\left[\mathbf{z}_t^r, \mathbf{z}_t^d\right] \in \mathbb{R}^{c \times f \times 2h \times w}. \quad (1)$$

The transformer backbone $\mathcal{F}_\theta$ takes tokenized multi-modal inputs and predicts the denoising direction in token space:

$$\mathbf{y}_t = \mathcal{F}_\theta(\mathbf{z}_t, t) \in \mathbb{R}^{N \times c}, \quad N = f \times 2h \times w. \quad (2)$$

The reverse update is then performed by a scheduler $\mathcal{S}$ (Song et al., 2020; Lipman et al., 2022):

$$\mathbf{z}_{t-1} = \mathcal{S}(\mathbf{z}_t, \mathbf{y}_t, t). \quad (3)$$

**Feature Caching for Diffusion Models.** Feature caching (Selvaraju et al., 2024; Liu et al., 2025a) accelerates sampling by reusing (or cheaply predicting) model outputs across denoising steps. A generic *model-level* caching scheme replaces the expensive backbone evaluation with a cached surrogate:

$$\tilde{\mathbf{y}}_t = \mathcal{C}_t(\mathbf{z}_t, t; \mathcal{H}_t), \quad (4)$$

where $\mathcal{H}_t$ stores information from previous FULL evaluations (e.g., past outputs $\mathbf{y}_{t'}$), and $\mathcal{C}_t$ specifies how cached computation is formed (e.g., direct reuse, interpolation, or lightweight prediction). Through $\mathcal{C}_t$, the expensive forward $\mathcal{F}_\theta(\cdot)$ is invoked only intermittently, enabling faster world-model sampling.

## 4. WorldCache

### 4.1. Curvature-guided Heterogeneous Token Prediction

**Observation ❶.** *World models exhibit strong token heterogeneity.* As shown in Fig. 3, world models mix heterogeneous modalities (e.g., RGB video vs. depth) and exhibit large spatial variance, yielding markedly different token trajectories across denoising steps. As a result, a *single global* caching rule (e.g., always reuse or always apply the same linear predictor) is mismatched: conservative rules waste computation on stable tokens, while aggressive rules fail on a small subset of chaotic tokens and cause global drift.

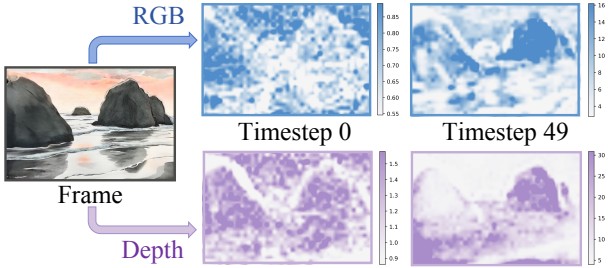

*(a)* Visualization of curvature maps for RGB and Depth modalities.

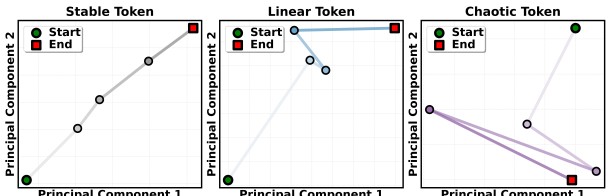

*(b)* PCA trajectory visualization of different type tokens.

*Figure 3.* **An illustration of token heterogeneity. (a) Modality and Spatial Variance:** Distinct patterns between modalities and across spatial regions. **(b) Trajectory Dynamics:** Three trajectory trends: static, predictable, and sharp, non-linear direction shifts that defy simple extrapolation. **More analysis in Appendix Sec. E**

This motivates a *token-adaptive* caching strategy: estimate how difficult each token is to predict from cached history, then apply different approximations accordingly, allocating compute only to tokens that need it.

**Curvature as a physics-grounded predictability cue.** To quantify *how predictable* each token is under caching, we measure the local nonlinearity of its temporal trajectory via a curvature-like score. Let $\mathbf{y}_t \in \mathbb{R}^{N \times d}$ be the FULL computation output in token space at timestep $t$, and let $\mathbf{y}_{t,i} \in \mathbb{R}^d$ denote token $i$ (the $i$-th row) of $\mathbf{y}_t$. Given the last three FULL outputs at timesteps $t_2 > t_1 > t_0$ (following the denoising order), we define discrete velocities

$$\mathbf{v}_{t_0,i} = \frac{\mathbf{y}_{t_0,i} - \mathbf{y}_{t_1,i}}{t_0 - t_1}, \qquad \mathbf{v}_{t_1,i} = \frac{\mathbf{y}_{t_1,i} - \mathbf{y}_{t_2,i}}{t_1 - t_2}, \quad (5)$$

and a discrete acceleration

$$\mathbf{a}_{t_0,i} = \frac{\mathbf{v}_{t_0,i} - \mathbf{v}_{t_1,i}}{t_0 - t_1}. \quad (6)$$

We compute the curvature score

$$\kappa_i = \frac{\|\mathbf{a}_{t_0,i}\|_2}{\|\mathbf{v}_{t_0,i}\|_2^2 + \varepsilon}, \quad (7)$$

where $\varepsilon$ is a small constant (e.g., $\varepsilon = 1e^{-8}$). This formulation is physically motivated (Federer, 1959): $\mathbf{v}$ captures the local drift of token features along denoising time, while $\mathbf{a}$ captures how quickly this drift changes. More importantly, the relevant skipped-step error is the deviation between the

true future token and its local first-order surrogate. For a smooth local trajectory $y_i(\tau)$, Taylor expansion gives

$$\|y_i(\tau + \Delta) - y_i(\tau) - \Delta y_i'(\tau)\|_2 \leq \frac{\Delta^2}{2} \sup_{\xi \in [\tau, \tau + \Delta]} \|y_i''(\xi)\|_2. \quad (8)$$

Hence, the cache difficulty of first-order prediction is governed by second-order departure from local linearity rather than raw displacement magnitude. In the smooth limit, curvature, namely second-order variation normalized by local speed squared, upper-bounds this local error under a local speed bound. Eq. (7) is its finite-difference counterpart estimated from the last three FULL outputs. Appendix Sec. A provides the formal statements and proofs. Thus, $\kappa_i$ acts as a normalized "turning rate." Small curvature indicates nearly linear evolution that is amenable to reuse or extrapolation. By contrast, large curvature indicates fast direction changes, where naive caching becomes more drift-prone.

**Token grouping by curvature.** We partition tokens by curvature percentiles (computed from $\{\kappa_i\}_{i=1}^N$):

$$\begin{aligned}
\mathcal{I}_{\text{stable}} &= \{i : \kappa_i < Q_{p_s}(\kappa)\}, \\
\mathcal{I}_{\text{chaotic}} &= \{i : \kappa_i \geq Q_{p_c}(\kappa)\}, \quad (9) \\
\mathcal{I}_{\text{linear}} &= [N] \setminus (\mathcal{I}_{\text{stable}} \cup \mathcal{I}_{\text{chaotic}}),
\end{aligned}$$

where $Q_p(\kappa)$ denotes the $p$-quantile and $(p_s, p_c)$ are fixed percentiles. The masks are refreshed whenever a new FULL output is obtained and three FULL outputs are available.

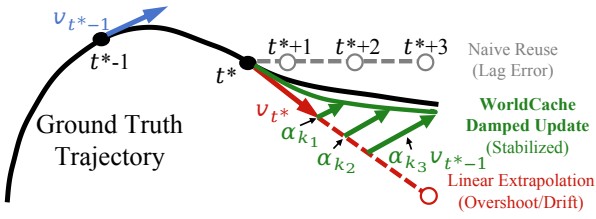

*(a)* Visualization of different update direction.

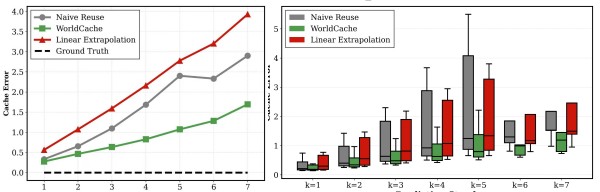

*(b)* Quantitative analysis of cache error under different updates.

*Figure 4.* **Mechanism and effectiveness of the Damped Update.** **(a) Trajectory Illustration:** Damped update stabilizes prediction through historical $\mathbf{v}_{t^\star-1}$. **(b) Quantitative Error Analysis:** Damped update reduces chaotic tokens cache error as the prediction window $k$ increases.

**Heterogeneous prediction: easy tokens vs. chaotic tokens.** Let $t^\star$ denote the most recent FULL computation timestep, and let $k$ be the number of consecutive CACHE

steps since $t^\star$. We denote the most recent FULL output as $\mathbf{y}_{t^\star}$, the corresponding velocity $\mathbf{v}_{t^\star}$, and its token as $\mathbf{y}_{t^\star,i}$. In CACHE, we construct a surrogate token-space output $\tilde{\mathbf{y}}_t$ token-wise:

$$\tilde{\mathbf{y}}_{t,i} = \begin{cases} \mathbf{y}_{t^\star,i}, & i \in \mathcal{I}_{\text{stable}}, \\ \mathbf{y}_{t^\star,i} + k \cdot \mathbf{v}_{t^\star,i}, & i \in \mathcal{I}_{\text{linear}}, \\ \mathbf{y}_{t^\star,i} + k \cdot \mathbf{v}_i^{\text{adapt}}(k), & i \in \mathcal{I}_{\text{chaotic}}. \end{cases} \quad (10)$$

Here $\mathbf{v}_{t^\star,i}$ is the latest cached velocity (computed from the two most recent FULL outputs).

**Chaotic group.** Tokens in $\mathcal{I}_{\text{chaotic}}$ exhibit high curvature with abrupt direction shifts; naive 1st-order extrapolation quickly accumulates errors and causes drift. To stabilize long cached streaks, we adopt a curvature-aware *damped* update by blending two recent velocities with a cubic Hermite (smoothstep) schedule (Weisstein, 2002):

$$\begin{aligned}
\mathbf{v}_i^{\text{adapt}}(k) &= (1 - \alpha_k)\mathbf{v}_{t^\star,i} + \alpha_k \mathbf{v}_{t^\star-1,i}, \\
\alpha_k &= 3x_k^2 - 2x_k^3, \quad x_k = \min\left(\frac{k}{n_{\max}}, 1\right),
\end{aligned} \quad (11)$$

where $n_{\max}$ is the maximum cached streak. As shown in Fig. 4, this design reduces reliance on a single-step tangent direction. As $k$ grows, $\alpha_k$ increases and the update becomes more conservative, mitigating drift under high-curvature dynamics while retaining caching efficiency.

### 4.2. Chaotic-prioritized Adaptive Skipping

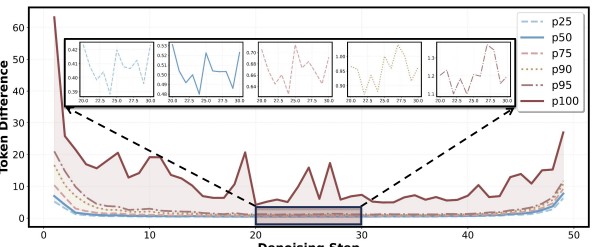

*Figure 5.* **An illustration of non-uniform temporal dynamics.** We plot the feature difference magnitude across denoising steps for different token percentiles ($p_{25}$ to $p_{100}$). The global drift is dominated by a small subset of "hard" tokens (top percentile, red line), while the majority remain stable. **More analysis in Appendix Sec. F.**

**Observation ❷.** *World models exhibit non-uniform temporal dynamics with token-dependent difficulty.* As shown in Fig. 5, timesteps in world-model denoising are not equally challenging: trajectories can be smooth for many steps and then abruptly become highly non-linear. Such temporal "hardness" is typically variant within tokens (only subsets of tokens vary much at a single timestep). Therefore, we should monitor *accumulated drift on the hardest tokens* and trigger FULL computation only when their uncertainty indicates imminent divergence.

**Dimensionless normalized drift.** A key goal of adaptive skipping is to compare *accumulated* deviations across timesteps under heterogeneous token statistics. However, raw feature differences (e.g., $\|\tilde{\mathbf{y}}_{t,i} - \tilde{\mathbf{y}}_{t+1,i}\|_2$) are scale-dependent: their magnitudes vary with modality-specific norms and timestep-dependent distribution shifts, making a unified threshold unreliable. We address this by building a dimensionless drift primitive using curvature.

**Theorem 4.1** (Curvature-induced dimensionless normalization). *Let $\kappa_i$ be computed by Eq. (7). For any feature deviation $\Delta\mathbf{y}_{t,i}$ that shares the same modality/timestep scalar units as $\mathbf{y}_{t,i}$ (e.g., $\Delta\mathbf{y}_{t,i} = \tilde{\mathbf{y}}_{t,i} - \tilde{\mathbf{y}}_{t+1,i}$), the product $\kappa_i \cdot \|\Delta\mathbf{y}_{t,i}\|_2$ is dimensionless in the sense that its leading dependence on global feature rescaling cancels: under $\mathbf{y} \mapsto \mathbf{y}' = s\mathbf{y}$ with $s > 0$,*

$$\kappa_i' \cdot \|\Delta\mathbf{y}_{t,i}'\|_2 = \kappa_i \cdot \|\Delta\mathbf{y}_{t,i}\|_2 + o(1), \qquad (12)$$

*where the residual $o(1)$ only arises from dimensionless numerical terms. **Detailed proofs in Appendix Sec. A***

Theorem 4.1 suggests a scale-normalized primitive for drift measurement: weight feature deviations by curvature to remove feature magnitude and make scores comparable across tokens and timesteps. And since curvature naturally quantifies the predictive hardness of each token, we only need to monitor the *chaotic* tokens for FULL computation.

For the chaotic token set $\mathcal{I}_{\text{chaotic}}$, we define the per-step normalized drift

$$e_i(t) = \kappa_i \cdot \|\tilde{\mathbf{y}}_{t,i} - \tilde{\mathbf{y}}_{t+1,i}\|_2, \qquad i \in \mathcal{I}_{\text{chaotic}}, \quad (13)$$

and aggregate it into a unified uncertainty score

$$E(t) = \frac{1}{|\mathcal{I}_{\text{chaotic}}|} \sum_{i \in \mathcal{I}_{\text{chaotic}}} e_i(t). \qquad (14)$$

Intuitively, $E(t)$ measures the *relative* drift (normalized by intrinsic trajectory scale) on the hardest tokens, and is thus robust to modality-dependent feature norms and timestep-wise distribution shifts.

**Accumulated uncertainty for FULL triggering.** We accumulate the normalized uncertainty over consecutive cached steps:

$$E_{\text{acc}} \leftarrow E_{\text{acc}} + E(t), \qquad (15)$$

and trigger a FULL backbone evaluation when $E_{\text{acc}}$ exceeds a single threshold $\eta$. Since each $E(t)$ is scale-normalized by Theorem 4.1, the same $\eta$ applies across timesteps and heterogeneous token distributions.

**4.3. Overall Framework**

---

**Algorithm 1** WorldCache Framework

**Require:** Initial latent $\mathbf{z}_T$; backbone $\mathcal{F}_\theta$; scheduler $\mathcal{S}$.
1: Init: history buffer $\mathcal{H} \leftarrow \emptyset$ (last 3 FULL outputs), masks $\mathcal{I}_{\text{stable}}, \mathcal{I}_{\text{linear}}, \mathcal{I}_{\text{chaotic}}$, counter $k \leftarrow 0$, accumulator $E_{\text{acc}} \leftarrow 0$, prev. surrogate $\tilde{\mathbf{y}}_{\text{prev}} \leftarrow \mathbf{0}$.
2: **for** $t = T, T-1, \ldots, 0$ **do**
3:     **if** $|\mathcal{H}| < 3$ **or** $E_{\text{acc}} \geq \eta$ **then**
4:         **FULL:** $\tilde{\mathbf{y}}_t \leftarrow \mathbf{y}_t = \mathcal{F}_\theta(\mathbf{z}_t, t)$.
5:         Update $\mathcal{H}$ with $(\mathbf{y}_t, t)$; if $|\mathcal{H}| = 3$, compute $\kappa$ and update masks (Eq. (7), (9)).
6:         Reset $k \leftarrow 0$, $E_{\text{acc}} \leftarrow 0$.
7:     **else**
8:         **CACHE:** $k \leftarrow k + 1$; predict $\tilde{\mathbf{y}}_t$ by heterogeneous token rules (reuse / linear / damped for chaotic, Eq. (11)).
9:         Compute $E(t)$ by Eq. (14) using $\tilde{\mathbf{y}}_t, \tilde{\mathbf{y}}_{\text{prev}}$; update $E_{\text{acc}} \leftarrow E_{\text{acc}} + E(t)$.
10:    **end if**
11:    $\mathbf{z}_{t-1} \leftarrow \mathcal{S}(\mathbf{z}_t, \tilde{\mathbf{y}}_t, t)$; $\tilde{\mathbf{y}}_{\text{prev}} \leftarrow \tilde{\mathbf{y}}_t$.
12: **end for**

---

**Pipeline summary.** WorldCache alternates between FULL and CACHE steps during denoising. In FULL, we evaluate the backbone to refresh cached outputs, estimate token curvature, and update token groups. In CACHE, we predict the surrogate output via heterogeneous token prediction (reuse / linear / damped) and update the curvature-normalized drift accumulator. When the accumulated normalized drift indicates imminent divergence of chaotic tokens, we switch back to FULL. Algorithm 1 summarizes the procedure.

## 5. Experiments

### 5.1. Experimental Settings

**Models.** We evaluate on two state-of-the-art multi-modal diffusion world models: HunyuanVoyager-13B (Huang et al., 2025) and Aether-5B (Zhu et al., 2025). Both take image, text, and camera trajectory as conditions, and produce coupled RGB video and depth outputs.

**Evaluation.** For world generation, we report two complementary types of metrics. **(i) WorldScore benchmark.** We use WorldScore (Duan et al., 2025), a comprehensive benchmark that evaluates world generation from both *controllability* and *quality*. **(ii) Perceptual consistency.** We measure the discrepancy to the no-cache baseline PSNR, SSIM (Wang et al., 2004), and LPIPS (Zhang et al., 2018)). For 3D reconstruction, we follow Aether (Zhu et al., 2025) and evaluate both depth and camera pose quality.

*Table 1.* **Quantitative comparison on Image-to-World generation. Bold**: the best result. *Note: Layer-wise methods marked with *
exceed single-GPU memory limits, requiring CPU offloading which incurs transmission latency.

| Method | Benchmark Evaluation | | Perceptual Metrics | | | Acceleration & Memory | | |
|---|---|---|---|---|---|---|---|---|
| | WorldScore Static↑ | WorldScore Dynamic↑ | PSNR↑ | SSIM↑ | LPIPS↓ | Latency(s)↓ | Speed↑ | Memory Overhead(GB)↓ |
| HunyuanVoyager-13B (Huang et al., 2025) ($512 \times 768p$, frames = 49) | | | | | | | | |
| Voyager | 66.28 | 46.40 | ∞ | 1.000 | 0.000 | 1053.7 | 1.00× | 50.44 |
| DuCa | 53.87 | 37.71 | 16.66 | 0.508 | 0.486 | 811.5* | 1.30× | 109.70 |
| ToCa | 47.49 | 33.24 | 15.51 | 0.409 | 0.558 | 1038.4* | 1.01× | 107.35 |
| TaylorSeer | 62.46 | 43.72 | 18.32 | 0.615 | 0.293 | 1195.2* | 0.88× | 163.79 |
| HiCache | 63.80 | 44.66 | 18.56 | 0.623 | 0.281 | 1100.1* | 0.96× | 163.79 |
| TeaCache | 60.88 | 42.61 | 16.25 | 0.565 | 0.372 | 311.5 | 3.38× | 56.52 |
| EasyCache | 64.16 | 44.91 | 21.76 | 0.737 | 0.208 | 294.5 | 3.58× | 50.98 |
| HERO | 62.37 | 43.67 | 17.71 | 0.601 | 0.315 | 1100 | 0.96× | 173.42 |
| **WorldCache** | **64.89** | **45.43** | **23.49** | **0.770** | **0.176** | **288.6** | **3.65×** | **50.58** |
| Aether-5B (Zhu et al., 2025) ($480 \times 720p$, frames = 41) | | | | | | | | |
| Aether | 64.60 | 45.22 | ∞ | 1.000 | 0.000 | 179.7 | 1.00× | 46.58 |
| DuCa | 60.17 | 42.12 | 26.68 | 0.838 | 0.151 | 110.3 | 1.63× | 61.44 |
| ToCa | 60.15 | 42.11 | 26.68 | 0.839 | 0.151 | 110.8 | 1.62× | 61.78 |
| TaylorSeer | 57.11 | 39.97 | 22.92 | 0.713 | 0.324 | 108.0 | 1.66× | 77.32 |
| HiCache | 58.96 | 41.27 | 24.93 | 0.784 | 0.226 | 108.8 | 1.65× | 77.32 |
| TeaCache | 60.95 | 42.67 | 26.60 | 0.843 | 0.138 | 114.2 | 1.57× | 46.78 |
| EasyCache | 62.89 | 44.02 | 22.84 | 0.720 | 0.186 | 120.9 | 1.49× | 46.59 |
| HERO | 58.62 | 41.04 | 23.56 | 0.741 | 0.259 | 132.0 | 1.36× | 75.08 |
| **WorldCache** | **63.68** | **44.72** | **31.87** | **0.924** | **0.066** | **107.2** | **1.68×** | **46.59** |

*Table 2.* Quantitative comparison on 3D Reconstruction on Aether (Zhu et al., 2025). **Bold**: the best result.

| Method | Video Depth Estimation | | | Camera Pose Estimation | | | Acceleration & Memory | | |
|---|---|---|---|---|---|---|---|---|---|
| | Abs Rel↓ | $\delta < 1.25$↑ | $\delta < 1.25^2$↑ | ATE↓ | RPE trans↓ | RPE rot↓ | Latency(s)↓ | Speed↑ | Memory Overhead(GB)↓ |
| Aether | 0.340 | 0.502 | 0.738 | 0.177 | 0.068 | 0.780 | 55.42 | 1.00× | 50.19 |
| DuCa | 0.341 | 0.475 | 0.694 | 0.209 | 0.069 | 0.904 | 28.15 | 1.97× | 52.70 |
| ToCa | 0.341 | 0.476 | 0.694 | 0.209 | 0.069 | 0.904 | 28.02 | 1.98× | 52.70 |
| TaylorSeer | 0.361 | 0.460 | 0.718 | 0.197 | 0.068 | 1.134 | 26.71 | 2.07× | 58.57 |
| HiCache | 0.346 | 0.472 | 0.712 | 0.204 | 0.069 | 1.004 | 26.51 | 2.09× | 58.57 |
| TeaCache | 0.341 | 0.496 | 0.724 | 0.183 | 0.068 | 0.797 | 25.85 | 2.14× | 50.20 |
| EasyCache | 0.390 | 0.479 | 0.725 | 0.183 | 0.069 | 1.061 | 27.76 | 2.00× | 50.20 |
| HERO | 0.347 | 0.490 | 0.716 | **0.181** | 0.071 | 0.861 | 27.44 | 1.96× | 61.56 |
| **WorldCache** | **0.341** | **0.508** | **0.741** | 0.184 | **0.068** | **0.796** | **21.20** | **2.61×** | **50.20** |

**Baselines.** We compare to representative training-free diffusion caching methods, including layer-wise caching (DuCa (Zou et al., 2024b), ToCa (Zou et al., 2024a), TaylorSeer (Liu et al., 2025b), HiCache (Feng et al., 2026a)), model-wise caching (TeaCache (Liu et al., 2025a), Easy-Cache (Zhou et al., 2025)), and HERO (Song et al., 2025) which combines caching with token merging.

All the experiments are conducted on a single NVIDIA-A800 GPU. **More detailed settings and descriptions are provided in Appendix Sec. B and Sec. C.**

## 5.2. World Generation Results

Tab. 1 summarizes image-to-world generation results on HunyuanVoyager-13B (Huang et al., 2025) and Aether-5B (Zhu et al., 2025). On **Voyager-13B**, WORLDCACHE attains the best perceptual metrics (PSNR 23.49 vs. 21.76 of EasyCache) and near-lossless WorldScore (45.43 compared to baseline 46.40), while achieving **3.65×** end-to-end acceleration with essentially no extra memory (50.58GB vs. 50.44GB baseline). Notably, layer-wise caching baselines incur substantial memory overhead (> 100GB), which can not fit in a single GPU yet do not reliably improve throughput and often degrade fidelity, highlighting their

mismatch to multi-modal world simulation. On **Aether-5B**, WORLDCACHE again yields the strongest fidelity with the best WorldScore among accelerated methods (44.72 vs. 44.02 of EasyCache) and the highest speedup (**1.68×**) under near-zero memory overhead. These results support our design: token-wise heterogeneous caching preserves difficult multi-modal regions, while chaotic-prioritized skipping prevents drift under non-uniform denoising dynamics. The additional control path has a negligible impact: curvature estimation, grouping and trigger evaluation account for only around 0.05% of end-to-end latency across models (see Appendix Sec. D). Therefore, the observed gains stem from cheaper cached prediction rather than control overhead.

### 5.3. 3D Reconstruction Results

Tab. 2 also reports 3D reconstruction results on Aether for depth and pose estimation. WorldCache preserves geometry-aware capability with near-lossless performance while providing the largest 2.61× acceleration. For **depth**, it matches the best Abs Rel (0.341 compared to baseline 0.340) and achieves the highest $\delta$ accuracy. For **pose**, it attains the lossless RPE trans (0.068) with the lowest rotation error among accelerated methods (0.796 compared to 0.861 of HERO). Meanwhile, WORLDCACHE reduces reconstruction latency to **21.20s** (**2.61×**), outperforming all baselines.

*Table 3.* Ablation study on token grouping percentiles.

| $\{p_s, p_c\}$ | PSNR↑ | SSIM↑ | LPIPS↓ |
|---|---|---|---|
| EasyCache | 21.76 | 0.737 | 0.208 |
| $\{0.3, 0.8\}$ | 23.32 | 0.766 | 0.179 |
| $\{0.3, 0.7\}$ | 23.49 | **0.770** | **0.176** |
| $\{0.3, 0.6\}$ | **23.52** | 0.768 | 0.178 |
| $\{0.2, 0.8\}$ | 23.12 | 0.760 | 0.185 |
| $\{0.2, 0.7\}$ | 23.12 | 0.764 | 0.182 |
| $\{0.2, 0.6\}$ | 23.33 | 0.764 | 0.181 |
| $\{0.1, 0.8\}$ | 22.77 | 0.758 | 0.188 |
| $\{0.1, 0.7\}$ | 22.97 | 0.758 | 0.185 |
| $\{0.1, 0.6\}$ | 22.95 | 0.758 | 0.187 |

### 5.4. Visual Comparison

Fig. 6 compares qualitative results on both HunyuanVoyager and Aether. Most baselines exhibit visible drift under caching, including high-frequency color noise or local blurring; these artifacts are especially evident around textured regions and boundaries (see insets). On Aether, errors also manifest in the coupled RGB–depth outputs as boundary bleeding and inconsistent depth regions. In contrast, WORLDCACHE produces results closest to the `Original` in both appearance and geometry, preserving sharper structures and cleaner depth maps, consistent with our token-adaptive caching and chaotic-prioritized skipping design. **We provide more visual comparisons in Appendix Sec. H.**

*Table 4.* Ablation study on adaptive skipping threshold ($\eta$).

| Method | PSNR↑ | SSIM↑ | LPIPS↓ | Latency(s)↓ |
|---|---|---|---|---|
| TeaCache | 26.60 | 0.843 | 0.138 | 114.2 |
| CAS ($\eta = 0.10$) | 31.66 | 0.922 | 0.070 | 109.0 |
| CAS ($\eta = 0.15$) | 31.13 | 0.916 | 0.083 | 108.5 |
| CAS ($\eta = 0.20$) | 30.63 | 0.908 | 0.099 | 107.2 |
| CAS ($\eta = 0.25$) | 29.22 | 0.907 | 0.133 | 99.35 |
| CAS ($\eta = 0.30$) | 28.05 | 0.894 | 0.154 | 93.86 |
| CAS ($\eta = 0.35$) | 27.10 | 0.881 | 0.198 | 90.35 |

### 5.5. Ablation Studies

We ablate the two core components of **WorldCache**: curvature-guided heterogeneous token prediction (CHTP) and chaotic-prioritized adaptive skipping (CAS).

**Token prediction strategies.** Tab. 5 provides further comparisons between the CHTP and the uniform and randomly grouped predictors. While uniform reuse is inexpensive, it cannot track evolving tokens. Uniform linear extrapolation performs poorly in regions of high curvature, and uniform damped updates introduce unnecessary historical bias on easy tokens. Mixing these operators at random is also insufficient, indicating that the improvement does not merely come from operator diversity. Curvature-based grouping, on the other hand, identifies which tokens should be reused, extrapolated or damped, yielding the best fidelity at nearly identical latency.

*Table 5.* Ablation study on token prediction methods.

| Method | PSNR↑ | SSIM↑ | LPIPS↓ | Latency(s)↓ |
|---|---|---|---|---|
| Reuse | 22.74 | 0.714 | 0.336 | **86.32** |
| Linear | 18.01 | 0.537 | 0.396 | 87.07 |
| Damped | 23.76 | 0.665 | 0.276 | 87.51 |
| Random Group. | 22.59 | 0.710 | 0.314 | 86.98 |
| **CHTP** | **25.76** | **0.791** | **0.227** | 86.94 |

**Grouping percentiles.** Tab. 3 evaluates the sensitivity of the grouping thresholds $p_s$ and $p_c$ in Eq. 9 on Voyager. Performance remains consistent over a wide range: all nine tested pairs surpass the most robust uniform prediction baseline, EasyCache (Zhou et al., 2025), with PSNR ranging from 22.77 to 23.52, SSIM from 0.758 to 0.770, and LPIPS from 0.176 to 0.188. This broad plateau demonstrates that the improvement is not driven by tuning, but rather primarily by the heterogeneous grouping principle itself, rather than by the selection of specific percentiles. We use the default setting of $\{p_s, p_c\} = \{0.3, 0.7\}$, which achieves the most balanced perceptual quality.

**Skipping strategies.** Tab. 6 compares CAS with several alternative triggers. Although global difference- or norm-guided policies outperform fixed schedules, their thresholds

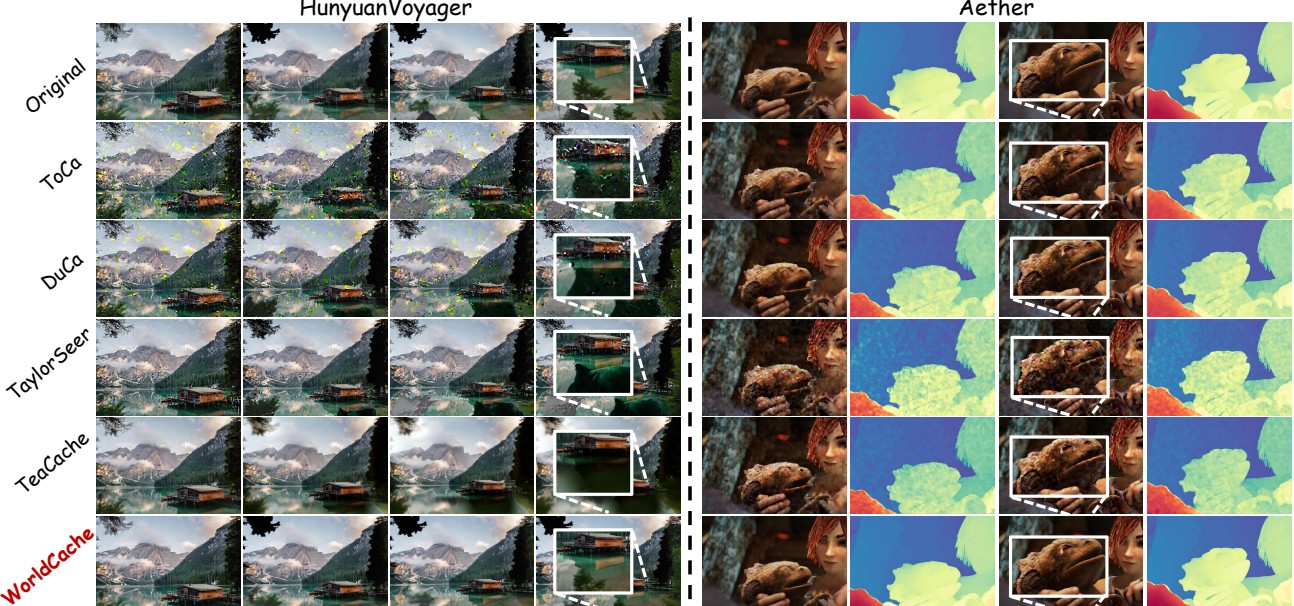

*Figure 6.* **Qualitative comparison of world generation tasks on Voyager (Huang et al., 2025) and Aether (Zhu et al., 2025).**

inherit large-scale variation across timesteps and modalities, making them difficult to calibrate consistently. Curvature-only triggering is also insufficient as hardness without actual displacement causes unnecessary recomputation. CAS performs better because it combines both difficulty and actual displacement in a dimensionless score, while only monitoring the chaotic subset and avoiding dilution by easy tokens.

*Table 6.* Ablation study on steps skipping strategies.

| Method | PSNR↑ | SSIM↑ | LPIPS↓ |
|---|---|---|---|
| Fixed Interval | 26.18 | 0.830 | 0.216 |
| Difference Guided | 26.79 | 0.824 | 0.207 |
| Norm Guided | 26.02 | 0.809 | 0.217 |
| Curvature Guided | 25.87 | 0.788 | 0.236 |
| **CAS** | **27.10** | **0.881** | **0.198** |

**Error threshold for adaptive skipping.** Tab. 4 evaluates the CAS triggering threshold $\eta$ in Eq. 15 on Aether. Smaller $\eta$ triggers FULL evaluations more frequently, leading to higher fidelity but slightly higher latency, while larger $\eta$ increases speed at the cost of visible drift. Across a wide range, CAS consistently improves over adaptive skipping baseline TeaCache (Liu et al., 2025a), demonstrating that our curvature-normalized drift provides a reliable skipping control and achieves both better quality and speed. We set $\eta = 0.20$ by default as a balanced operating point.

**We provide more analysis of different token prediction strategies in Appendix Sec. E and of adaptive skipping metrics in Appendix Sec. F.**

## 6. Conclusion

In this paper, we presented WorldCache, a training-free acceleration framework tailored for multi-modal diffusion world models. We identified that prior caching methods fail to address *token heterogeneity* arising from complex multi-modal and spatial dynamics. To resolve this, we introduced *Curvature-guided Heterogeneous Token Prediction*, which assigns physics-grounded strategies based on feature non-linearity, and *Chaotic-prioritized Adaptive Skipping*, which shifts the update paradigm from global averaging to a bottleneck-driven mechanism. Extensive experiments demonstrate that WorldCache achieves up to $3.7\times$ acceleration while preserving 98% of generative quality, offering a partial solution for efficient, interactive world simulation.

## Acknowledgements

This work is supported by the National Natural Science Foundation of China under Grant Number 62406312 and 62476264, the Beijing Natural Science Foundation under Grant Number 4244098, the Science Foundation of the Chinese Academy of Sciences, and the Swiss National Science Foundation (SNSF) project 200021E_219943 Neuromorphic Attention Models for Event Data (NAMED).

## Impact Statement

This paper presents work whose goal is to advance the field of machine learning. There are many potential societal consequences of our work, none of which we feel must be specifically highlighted here.

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

## A. Additional Theoretical Justification for Curvature-guided Caching

We use $y_i : [\tau, \tau + \Delta] \to \mathbb{R}^d$ to denote the local continuous trajectory of token $i$, and view the discrete FULL outputs $\mathbf{y}_{t_j,i}$ as samples of this trajectory at nearby denoising steps. This lets us state the approximation properties of reuse, linear extrapolation, curvature scoring, and hard-token monitoring in a unified notation.

**Lemma A.1** (Local truncation error of skipped-step predictors). *Let $y_i : [\tau, \tau + \Delta] \to \mathbb{R}^d$ denote the local trajectory of token $i$ over a skipped interval of length $\Delta > 0$.*

*1. If $y_i \in C^1([\tau, \tau + \Delta])$, the reuse predictor $\hat{y}_i^{\text{reuse}}(\tau + \Delta) = y_i(\tau)$ satisfies*

$$\|y_i(\tau + \Delta) - \hat{y}_i^{\text{reuse}}(\tau + \Delta)\|_2 \leq \Delta \sup_{\xi \in [\tau, \tau + \Delta]} \|y_i'(\xi)\|_2. \tag{16}$$

*2. If $y_i \in C^2([\tau, \tau + \Delta])$, the first-order linear predictor $\hat{y}_i^{\text{lin}}(\tau + \Delta) = y_i(\tau) + \Delta y_i'(\tau)$ satisfies*

$$\|y_i(\tau + \Delta) - \hat{y}_i^{\text{lin}}(\tau + \Delta)\|_2 \leq \frac{\Delta^2}{2} \sup_{\xi \in [\tau, \tau + \Delta]} \|y_i''(\xi)\|_2. \tag{17}$$

*Proof.* For reuse, apply the fundamental theorem of calculus:

$$y_i(\tau + \Delta) - y_i(\tau) = \int_\tau^{\tau + \Delta} y_i'(\xi) \, d\xi, \tag{18}$$

then take norms and upper-bound the integrand by its supremum on the interval. For the linear predictor, Taylor's theorem with remainder gives

$$y_i(\tau + \Delta) = y_i(\tau) + \Delta y_i'(\tau) + \frac{\Delta^2}{2} y_i''(\xi) \tag{19}$$

for some $\xi \in [\tau, \tau + \Delta]$, which immediately yields Eq. (17). □

**Lemma A.2** (Curvature upper-bounds first-order cache difficulty). *Assume $y_i \in C^2([\tau, \tau + \Delta])$, and define the regularized smooth-limit curvature score*

$$\kappa_i^{(\varepsilon)}(\xi) := \frac{\|y_i''(\xi)\|_2}{\|y_i'(\xi)\|_2^2 + \varepsilon}, \qquad \varepsilon > 0. \tag{20}$$

*If the token speed is locally bounded by $\|y_i'(\xi)\|_2 \leq v_{\max}$ for all $\xi \in [\tau, \tau + \Delta]$, then*

$$\|y_i(\tau + \Delta) - \hat{y}_i^{\text{lin}}(\tau + \Delta)\|_2 \leq \frac{\Delta^2}{2} (v_{\max}^2 + \varepsilon) \sup_{\xi \in [\tau, \tau + \Delta]} \kappa_i^{(\varepsilon)}(\xi). \tag{21}$$

*Proof.* By definition,

$$\|y_i''(\xi)\|_2 = \kappa_i^{(\varepsilon)}(\xi) \big(\|y_i'(\xi)\|_2^2 + \varepsilon\big) \leq \kappa_i^{(\varepsilon)}(\xi)(v_{\max}^2 + \varepsilon). \tag{22}$$

Substituting this into Eq. (17) gives Eq. (21). □

**Proposition A.3** (Consistency of the discrete curvature score). *Let $y_i \in C^3([\tau - 2h, \tau])$ for some $h > 0$, and define the backward finite differences*

$$v_{h,i}(\tau) = \frac{y_i(\tau) - y_i(\tau - h)}{h}, \qquad a_{h,i}(\tau) = \frac{v_{h,i}(\tau) - v_{h,i}(\tau - h)}{h}, \tag{23}$$

*with discrete regularized curvature*

$$\kappa_{h,i}^{(\varepsilon)}(\tau) := \frac{\|a_{h,i}(\tau)\|_2}{\|v_{h,i}(\tau)\|_2^2 + \varepsilon}. \tag{24}$$

*Also define the smooth-limit quantity*

$$\kappa_i^{(\varepsilon)}(\tau) := \frac{\|y_i''(\tau)\|_2}{\|y_i'(\tau)\|_2^2 + \varepsilon}. \tag{25}$$

*If $\|y_i'(\tau)\|_2^2 + \varepsilon$ is bounded away from zero, then*

$$\kappa_{h,i}^{(\varepsilon)}(\tau) = \kappa_i^{(\varepsilon)}(\tau) + O(h). \tag{26}$$

*Proof.* Taylor expansion around $\tau$ yields

$$v_{h,i}(\tau) = y'_i(\tau) - \frac{h}{2}y''_i(\tau) + O(h^2), \tag{27}$$

and, after expanding $y_i(\tau - h)$ and $y_i(\tau - 2h)$ around $\tau$,

$$v_{h,i}(\tau - h) = y'_i(\tau) - \frac{3h}{2}y''_i(\tau) + O(h^2). \tag{28}$$

Therefore,

$$a_{h,i}(\tau) = \frac{v_{h,i}(\tau) - v_{h,i}(\tau - h)}{h} = y''_i(\tau) + O(h), \tag{29}$$

while $\|v_{h,i}(\tau)\|_2^2 = \|y'_i(\tau)\|_2^2 + O(h)$. Substituting these two expansions into the definition of $\kappa^{(\varepsilon)}_{h,i}(\tau)$ proves first-order consistency. $\qquad\square$

**Proposition A.4** (Hard-token dilution under global monitoring). *Let $e_i(t) \geq 0$ denote token-wise cache error at denoising step $t$, and let $\mathcal{H} \subset [N]$ be a hard-token subset of size $m := |\mathcal{H}|$, with complement $\mathcal{E} = [N] \setminus \mathcal{H}$. Define*

$$E_{\mathcal{H}}(t) := \frac{1}{m}\sum_{i \in \mathcal{H}} e_i(t), \qquad E_{\mathcal{E}}(t) := \frac{1}{N-m}\sum_{i \in \mathcal{E}} e_i(t), \tag{30}$$

*and the global average*

$$E_{\text{all}}(t) := \frac{1}{N}\sum_{i=1}^{N} e_i(t). \tag{31}$$

*Then*

$$E_{\text{all}}(t) = \frac{m}{N}E_{\mathcal{H}}(t) + \left(1 - \frac{m}{N}\right)E_{\mathcal{E}}(t). \tag{32}$$

*Consequently, if $E_{\mathcal{E}}(t) \leq \epsilon$ while failure is governed by the hard subset, then any global trigger $E_{\text{all}}(t) \geq \tau$ cannot fire before*

$$E_{\mathcal{H}}(t) \geq \frac{N}{m}\left(\tau - \left(1 - \frac{m}{N}\right)\epsilon\right). \tag{33}$$

*Proof.* Eq. (32) is obtained by decomposing the global sum into the hard subset and its complement. The threshold condition follows by solving Eq. (32) for $E_{\mathcal{H}}(t)$ under the assumption $E_{\mathcal{E}}(t) \leq \epsilon$. $\qquad\square$

**Theorem A.5** (Curvature-induced dimensionless normalization). *Let $\kappa_i$ be computed by Eq. (7). For any feature deviation $\Delta\mathbf{y}_{t,i}$ that shares the same modality/timestep units as $\mathbf{y}_{t,i}$, the product $\kappa_i \cdot \|\Delta\mathbf{y}_{t,i}\|_2$ is dimensionless in the sense that its leading dependence on global feature rescaling cancels: under $\mathbf{y} \mapsto \mathbf{y}' = s\mathbf{y}$ with $s > 0$,*

$$\kappa'_i \cdot \|\Delta\mathbf{y}'_{t,i}\|_2 = \kappa_i \cdot \|\Delta\mathbf{y}_{t,i}\|_2 + o(1), \tag{34}$$

*where the residual $o(1)$ only arises from dimensionless numerical/regularization terms.*

*Proof.* Fix token index $i$ and omit it when clear. Recall the discrete definitions (Eq. (7)) using three FULL outputs at timesteps $t_0 > t_1 > t_2$:

$$\mathbf{v}_{t_0} = \frac{\mathbf{y}_{t_0} - \mathbf{y}_{t_1}}{t_0 - t_1}, \qquad \mathbf{v}_{t_1} = \frac{\mathbf{y}_{t_1} - \mathbf{y}_{t_2}}{t_1 - t_2}, \qquad \mathbf{a}_{t_0} = \frac{\mathbf{v}_{t_0} - \mathbf{v}_{t_1}}{t_0 - t_1}, \tag{35}$$

and the curvature score

$$\kappa = \frac{\|\mathbf{a}_{t_0}\|_2}{\|\mathbf{v}_{t_0}\|_2^2 + \varepsilon}. \tag{36}$$

**Effect of global feature rescaling.** Consider $\mathbf{y} \mapsto \mathbf{y}' = s\mathbf{y}$ with $s > 0$. Because finite differences are linear in $\mathbf{y}$,

$$\mathbf{v}'_{t_0} = s\mathbf{v}_{t_0}, \qquad \mathbf{v}'_{t_1} = s\mathbf{v}_{t_1}, \qquad \mathbf{a}'_{t_0} = s\mathbf{a}_{t_0}. \tag{37}$$

Substituting into Eq. (36) yields

$$\kappa' = \frac{\|\mathbf{a}'_{t_0}\|_2}{\|\mathbf{v}'_{t_0}\|_2^2 + \varepsilon} = \frac{s\|\mathbf{a}_{t_0}\|_2}{s^2\|\mathbf{v}_{t_0}\|_2^2 + \varepsilon} = \frac{1}{s} \cdot \kappa \cdot \frac{\|\mathbf{v}_{t_0}\|_2^2 + \varepsilon}{\|\mathbf{v}_{t_0}\|_2^2 + \varepsilon/s^2}. \tag{38}$$

The multiplicative ratio in Eq. (38) is dimensionless and approaches 1 whenever $\varepsilon$ is negligible compared to $\|\mathbf{v}_{t_0}\|_2^2$. Hence

$$\kappa' = \frac{1}{s}\kappa + o\left(\frac{1}{s}\right). \tag{39}$$

**Effect on the deviation norm.** For any deviation $\Delta\mathbf{y}$ in the same feature space, the same rescaling gives $\Delta\mathbf{y}' = s\,\Delta\mathbf{y}$ and therefore

$$\|\Delta\mathbf{y}'\|_2 = s\|\Delta\mathbf{y}\|_2. \tag{40}$$

**Cancellation in the product.** Combining Eq. (39) and Eq. (40),

$$\kappa' \cdot \|\Delta\mathbf{y}'\|_2 = \left(\frac{1}{s}\kappa + o\left(\frac{1}{s}\right)\right) \cdot (s\|\Delta\mathbf{y}\|_2) = \kappa \cdot \|\Delta\mathbf{y}\|_2 + o(1). \tag{41}$$

Therefore, the leading dependence on the global scale factor cancels, and the remaining discrepancy is induced only by dimensionless numerical/regularization terms. $\square$

## B. Detailed Description of Selected Evaluation Metrics

### B.1. WorldScore Metrics (Static & Dynamic)

WorldScore provides two overall scores: **WorldScore-Static** and **WorldScore-Dynamic**. It decomposes "world generation capability" into three aspects: *controllability*, *quality*, and *dynamics*. Each aspect consists of several individual metrics. After computing each metric in its raw form (some are errors where lower is better), we normalize every metric to a unified score in $[0, 100]$ so that *higher is always better*, and then aggregate them into overall scores.

**WorldScore-Static / WorldScore-Dynamic (Overall Scores).** Let $\{s_k\}$ denote the normalized scores (in $[0, 100]$) of individual metrics. **WorldScore-Static** measures *static world generation capability* by averaging the scores in **controllability** and **quality**:

$$\text{WorldScore-Static} = \frac{1}{7} \sum_{k \in \mathcal{K}_{\text{ctrl}} \cup \mathcal{K}_{\text{qual}}} s_k, \tag{42}$$

where $\mathcal{K}_{\text{ctrl}} = \{\text{Camera}, \text{Object}, \text{Content}\}$ and $\mathcal{K}_{\text{qual}} = \{\text{3D}, \text{Photo}, \text{Style}, \text{Subjective}\}$. **WorldScore-Dynamic** further evaluates *dynamic world generation capability* by incorporating three dynamics metrics:

$$\text{WorldScore-Dynamic} = \frac{1}{10} \sum_{k \in \mathcal{K}_{\text{ctrl}} \cup \mathcal{K}_{\text{qual}} \cup \mathcal{K}_{\text{dyn}}} s_k, \tag{43}$$

where $\mathcal{K}_{\text{dyn}} = \{\text{MotionAcc}, \text{MotionMag}, \text{MotionSmooth}\}$. For models that do not support dynamic tasks, the dynamics scores are set to 0.

**Controllability Metrics.** These metrics evaluate whether the model follows the *world specification* (camera/layout and text prompt).

- **Camera Control** measures how well the generated video follows a predefined camera trajectory. We estimate per-frame camera poses and compute a *rotation error* $e_\theta$ (in degrees) and a *scale-invariant* translation error $e_t$; they are combined by the geometric mean:

$$e_{\text{camera}} = \sqrt{e_\theta \cdot e_t}. \tag{44}$$

  The final camera control error is averaged over all frames and all test videos, and then mapped to a score in $[0, 100]$ (higher is better).

- **Object Control** evaluates whether the key objects specified in the next-scene prompt appear in the generated scene. We extract one or two object descriptions from the prompt and compute the *success rate* of open-set object detection by matching detected objects to these descriptions.

- **Content Alignment** measures whether the generated scene is aligned with the *entire* next-scene prompt (not only the detected objects). We compute a CLIP-based image–text consistency score (CLIPScore) between the prompt and the generated content, and then normalize it to $[0, 100]$.

**Quality Metrics.** These metrics focus on cross-frame coherence and perceptual quality in static worlds.

- **3D Consistency** measures geometric stability across frames. We reconstruct dense depth and camera poses and compute the *reprojection error* between co-visible pixels in consecutive frames; lower reprojection error indicates better 3D consistency, which is then mapped to a higher normalized score.

- **Photometric Consistency** measures appearance stability (e.g., texture flickering or identity/texture shifts) across frames. We estimate forward/backward optical flows between consecutive frames, track points forward and then back, and compute an Average End-Point Error (AEPE)-style deviation. A smaller deviation indicates stronger photometric consistency and yields a higher normalized score.

- **Style Consistency** measures whether the overall visual style drifts over time. We compute the Frobenius norm of the difference between the Gram matrices of the *first* and *last* frames in each next-scene generation task; smaller style drift corresponds to higher normalized score.

- **Subjective Quality** reflects human-perceived visual quality of generated scenes. It is computed by combining an image quality predictor (CLIP-IQA+) and an aesthetic predictor (CLIP-Aesthetic), and then normalized to $[0, 100]$.

**Dynamics Metrics (used only in WorldScore-Dynamic).** These metrics evaluate whether the model can generate motion that is *correctly placed*, *sufficiently strong*, and *temporally smooth*.

- **Motion Accuracy** measures whether motion happens in the intended regions (dynamic objects) rather than elsewhere. Given optical flow magnitude map $\mathbf{F}$ and a dynamic-object mask $\mathbf{M}$, we score motion placement by contrasting in-mask vs. out-of-mask motion magnitude.

- **Motion Magnitude** measures the overall strength of motion. It uses the median of the optical flow magnitude map $\mathbf{F}$ and averages it across frame pairs and videos.

- **Motion Smoothness** measures temporal stability of motion. We drop odd frames, reconstruct them using a video frame interpolation model, and compare reconstructed vs. original frames using MSE, SSIM, and LPIPS; their normalized results are averaged to produce the final smoothness score.

**Score Normalization (How per-metric values become comparable).** Because different metrics have different units and directions (some are errors, some are similarities), each raw metric is first mapped to $[0, 1]$ via empirical bounds (with a "higher-is-better" convention after mapping), clipped to $[0, 1]$, and finally scaled to $[0, 100]$ before aggregation.

### B.2. Perceptual Fidelity Metrics

Cache acceleration may introduce approximation errors (e.g., feature reuse / skipping), thus we additionally measure frame-level fidelity. For each prompt, we treat the *original (non-accelerated)* model output as the reference, and compute the following metrics between the reference video and the cache-accelerated video:

- **PSNR ($\uparrow$).** Peak signal-to-noise ratio computed per frame and then averaged over all frames and all prompts.

- **SSIM ($\uparrow$).** Structural similarity index (Wang et al., 2004) computed per frame and then averaged over all frames and all prompts.

- **LPIPS ($\downarrow$).** Learned perceptual image patch similarity (Zhang et al., 2018) computed per frame and then averaged (lower indicates closer perceptual similarity to the reference output).

## B.3. Acceleration & Memory Metrics

To quantify the practical benefits of cache acceleration, we report compute, runtime, and memory statistics:

- **FLOPs (T) ($\downarrow$).** Theoretical total floating-point operations for generating one full scene (reported in tera-FLOPs). We use a consistent counting protocol across methods and include all denoising (and other generation-critical) network computations.

- **Latency (s) ($\downarrow$).** End-to-end DiT inference time to generate one scene under a fixed hardware/software setup. We measure latency with warm-up runs and report the average over multiple trials.

- **Speed ($\uparrow$).** Speedup ratio defined as

$$\text{Speed} = \frac{\text{Latency(FP / vanilla)}}{\text{Latency(method)}}.$$

- **Memory Overhead (GB) ($\downarrow$).** Peak GPU memory of original model and the memory introduced by the cache mechanism (e.g., storing intermediate features / states), reported in gigabytes.

All efficiency and memory results are measured under identical batch size, resolution, number of frames, and inference hyperparameters to ensure a fair comparison across methods.

## B.4. 3D Reconstruction Metrics

Cache acceleration may disturb geometry-critical tokens and accumulate temporal drift, we additionally evaluate whether the accelerated world model preserves 3D-related capability. Following the reconstruction protocol of Aether (Zhu et al., 2025), we assess two tasks: depth estimation and camera pose estimation by comparing cache-accelerated predictions against the corresponding ground truth:

- **Abs Rel ($\downarrow$).** Absolute relative depth error computed per frame and then averaged over all frames and all samples.

- **$\delta < 1.25$ ($\uparrow$).** The fraction of pixels whose predicted depth is within $1.25\times$ of the ground-truth depth, computed per frame and then averaged.

- **$\delta < 1.25^2$ ($\uparrow$).** Similar to $\delta < 1.25$.

- **ATE ($\downarrow$).** Absolute trajectory error for camera poses after global Sim(3) alignment, measuring overall trajectory consistency.

- **RPE Trans ($\downarrow$).** Relative pose error in translation, measuring frame-to-frame translation drift.

- **RPE Rot ($\downarrow$).** Relative pose error in rotation, measuring frame-to-frame rotational drift.

# C. Detailed Experimental Settings

## C.1. Models and Inference Protocols

**HunyuanVoyager-13B.** We follow the standard inference protocol to generate $512 \times 768$p content with 49 frames using 50 denoising steps. Unless otherwise specified, all acceleration methods use the same scheduler and conditioning inputs as the official implementation.

**Aether-5B.** For world generation, we use the default 50-step inference to produce $480 \times 720$p content with 41 frames. For reconstruction, we adopt the 30-step setting as in Aether (Zhu et al., 2025). All methods share identical input conditions and schedulers for fair comparison.

For both models, we set $p_s = 0.3$, $p_c = 0.7$, $n_{\max} = 6$. For HunyuanVoyager (Huang et al., 2025), we set $\eta = 1.0$ and for Aether, we set $\eta = 0.2$. All the experiments are conducted on a single NVIDIA-A800 GPU.

## C.2. Evaluation Protocols and Metrics

**WorldScore benchmark.** We use WorldScore (Duan et al., 2025), which evaluates world generation from both *controllability* and *quality*. Following the benchmark protocol, inputs cover diverse indoor/outdoor scenarios with realistic and stylized conditions. We uniformly sample 40 single-scene prompts and 10 three-scene prompts across categories.

**Perceptual consistency to the no-cache baseline.** To measure fidelity of cached sampling relative to the original model behavior, we compute frame-level perceptual metrics between cached outputs and the corresponding no-cache baseline outputs: PSNR, SSIM (Wang et al., 2004), and LPIPS (Zhang et al., 2018). We report averaged metrics over the evaluated samples. All samples are generated with the same prompts used for the WorldScore benchmark.

**3D reconstruction metrics.** We follow Aether (Zhu et al., 2025) and HERO (Song et al., 2025) settings to use Sintel (Butler et al., 2012) dataset. We also follow the same experimental setting used in HERO.

## C.3. Baselines and Categorization

**Layer-wise caching.** These methods cache intermediate representations inside Transformer blocks, providing fine-grained reuse but typically introducing non-trivial memory overhead. We include DuCa (Zou et al., 2024b), ToCa (Zou et al., 2024a), TaylorSeer (Liu et al., 2025b), and HiCache (Feng et al., 2026a).

**Model-wise caching.** These methods cache at the model-output level and introduce minimal extra memory, which is favorable for multi-modal world models with heavy representations. We include TeaCache (Liu et al., 2025a) and EasyCache (Zhou et al., 2025). Our WORLDCACHE also belongs to this category.

**World-model-specific acceleration.** We additionally compare with HERO (Song et al., 2025), which combines caching with token merging.

# D. Broader Evaluation Beyond Main-text Settings

To test whether the gains in the main text transfer beyond the default Voyager/Aether configuration, we further evaluate WorldCache along four additional axes: an extra open-source world model, higher resolution, different denoising schedules, and longer rollouts. Across all these settings, WorldCache continues to preserve the best quality-efficiency trade-off over both model-wise and layer-wise baselines.

## D.1. Additional World Model and Resolutions

We first evaluate LingBot-14B (Team et al., 2026), a third diffusion world model, under the 49-frame, 70-step setting at two resolutions. Table 7 shows that the advantage of WorldCache persists on this new backbone. At $464 \times 832$, it improves over EasyCache by $+2.96$ PSNR while increasing speed from $2.16\times$ to $2.41\times$. At $720 \times 1280$, it again achieves the best fidelity and raises speed from $2.25\times$ to $2.51\times$, suggesting that heterogeneous token caching is not tied to a single architecture or resolution regime.

## D.2. Different Denoising Schedules and Longer Rollouts

We next vary the denoising horizon on HunyuanVoyager. As shown in Table 8, WorldCache remains the strongest method under both denser 70-step sampling and more aggressive 30-step sampling, indicating that the cache policy is not specialized to one diffusion schedule. We also extend LingBot rollouts beyond the 49-frame setting in Table 7. Table 9 shows that WorldCache continues to outperform existing baselines at 81 and 161 frames, with stable or slightly increasing speedup as the context grows.

## D.3. Runtime Overhead and Memory Behavior

The additional control path of WorldCache is negligible relative to end-to-end inference. As shown in Table 10, curvature estimation, percentile grouping, and trigger evaluation cost only 0.064–0.189s across models, about 0.05% of total latency. This explains why the observed speedups come from cheaper cached prediction rather than from shifting cost to the control logic. Table 11 further shows that the memory overhead of the model-level cache remains almost constant as context length

*Table 7.* Broader evaluation on LingBot-14B across two resolutions.

| Method | Resolution | PSNR↑ | SSIM↑ | LPIPS↓ | Speed↑ |
|---|---|---|---|---|---|
| DuCa | 464 × 832 | 19.10 | 0.716 | 0.182 | 1.87× |
| ToCa | 464 × 832 | 18.85 | 0.704 | 0.198 | 1.84× |
| TaylorSeer | 464 × 832 | 17.05 | 0.641 | 0.286 | 1.82× |
| HiCache | 464 × 832 | 17.82 | 0.678 | 0.241 | 1.86× |
| TeaCache | 464 × 832 | 19.85 | 0.742 | 0.136 | 2.08× |
| EasyCache | 464 × 832 | 20.28 | 0.751 | 0.118 | 2.16× |
| **WorldCache** | 464 × 832 | **23.24** | **0.812** | **0.079** | **2.41×** |
| DuCa | 720 × 1280 | 19.71 | 0.727 | 0.012 | 1.22× |
| ToCa | 720 × 1280 | 19.45 | 0.714 | 0.013 | 1.12× |
| TaylorSeer | 720 × 1280 | 17.60 | 0.651 | 0.018 | 0.86× |
| HiCache | 720 × 1280 | 18.39 | 0.688 | 0.015 | 0.95× |
| TeaCache | 720 × 1280 | 20.49 | 0.753 | 0.009 | 2.17× |
| EasyCache | 720 × 1280 | 20.93 | 0.762 | 0.007 | 2.25× |
| **WorldCache** | 720 × 1280 | **23.99** | **0.824** | **0.005** | **2.51×** |

*Table 8.* Broader evaluation on Voyager under different denoising schedules.

| Method | Steps | PSNR↑ | SSIM↑ | LPIPS↓ | Speed↑ |
|---|---|---|---|---|---|
| DuCa | 70 | 18.16 | 0.552 | 0.452 | 1.41× |
| ToCa | 70 | 17.01 | 0.453 | 0.524 | 1.10× |
| TaylorSeer | 70 | 19.82 | 0.659 | 0.259 | 0.98× |
| HiCache | 70 | 20.06 | 0.667 | 0.247 | 1.07× |
| TeaCache | 70 | 18.05 | 0.617 | 0.331 | 3.82× |
| EasyCache | 70 | 23.56 | 0.789 | 0.167 | 4.03× |
| **WorldCache** | 70 | **25.49** | **0.828** | **0.130** | **4.09×** |
| DuCa | 30 | 16.13 | 0.501 | 0.509 | 1.00× |
| ToCa | 30 | 14.98 | 0.402 | 0.581 | 0.78× |
| TaylorSeer | 30 | 17.79 | 0.608 | 0.316 | 0.66× |
| HiCache | 30 | 18.03 | 0.616 | 0.304 | 0.71× |
| TeaCache | 30 | 15.61 | 0.556 | 0.399 | 2.46× |
| EasyCache | 30 | 21.12 | 0.728 | 0.235 | 2.61× |
| **WorldCache** | 30 | **22.78** | **0.760** | **0.206** | **2.72×** |

grows, because WorldCache stores only a fixed-depth output history and token masks instead of per-layer activations. In

*Table 9.* Broader evaluation on longer LingBot rollouts beyond the 49-frame main setting.

| Method | Frames | PSNR↑ | SSIM↑ | LPIPS↓ | Speed↑ |
|--------|--------|-------|-------|--------|--------|
| ToCa | 81 | 19.54 | 0.666 | 0.137 | 1.03× |
| TaylorSeer | 81 | 17.32 | 0.608 | 0.212 | 0.97× |
| HiCache | 81 | 18.23 | 0.643 | 0.171 | 0.99× |
| TeaCache | 81 | 20.89 | 0.701 | 0.085 | 2.23× |
| EasyCache | 81 | 21.81 | 0.705 | 0.082 | 2.29× |
| **WorldCache** | 81 | **23.37** | **0.780** | **0.001** | **2.47×** |
| ToCa | 161 | 18.81 | 0.709 | 0.157 | 1.01× |
| TaylorSeer | 161 | 16.23 | 0.636 | 0.217 | 0.89× |
| HiCache | 161 | 17.26 | 0.676 | 0.182 | 0.93× |
| TeaCache | 161 | 20.47 | 0.756 | 0.118 | 2.42× |
| EasyCache | 161 | 21.81 | 0.777 | 0.133 | 2.45× |
| **WorldCache** | 161 | **22.16** | **0.803** | **0.001** | **2.53×** |

practice, the added memory stays at 0.02–0.03 GB from 49 to 161 frames.

*Table 10.* Runtime overhead of WorldCache control logic.

| Model | WorldCache Overhead (s)↓ | End-to-End Latency (s)↓ | Overhead / Latency↓ |
|-------|--------------------------|-------------------------|---------------------|
| LingBot-14B | 0.189 | 348.7 | 0.054% |
| Voyager-13B | 0.137 | 288.6 | 0.047% |
| Aether-5B | 0.064 | 107.2 | 0.060% |

*Table 11.* Memory behavior of WorldCache as context length increases.

| Context Length | Baseline Mem.(GB)↓ | WorldCache Mem.(GB)↓ | Extra Mem.(GB)↓ | Speed↑ |
|----------------|--------------------|----------------------|------------------|--------|
| 49 | 49.37 | 49.39 | 0.02 | 2.41× |
| 81 | 55.87 | 55.89 | 0.02 | 2.47× |
| 161 | 71.32 | 71.35 | 0.03 | 2.53× |

Degradation is most likely in highly dynamic regimes with rapid camera motion, complex textures, or abrupt geometric changes. In these cases, the error is dominated by a small chaotic-token subset, so aggressive caching may either trigger more frequent FULL recomputation and shrink the speedup, or, if the threshold is too loose, introduce visible local drift. CAS alleviates this trade-off by monitoring the hard subset directly, but it does not eliminate it entirely.

## E. More Analysis of Curvature-guided Heterogeneous Token Prediction

In this section, we provide further empirical evidence to substantiate the design choices of our *Curvature-guided Heterogeneous Token Prediction* (CHTP). We first visualize the pervasiveness of token heterogeneity across different prompts and

modalities, and then present a quantitative ablation study to validate the effectiveness of our grouping strategy.

## E.1. Visualization of Token Heterogeneity

To demonstrate that token heterogeneity is a widespread phenomenon in world models rather than an isolated case, we extend our visualization to multiple diverse prompts using HunyuanVoyager (Huang et al., 2025). As shown in Figure 7, we map the curvature $\kappa$ of both RGB and Depth tokens across various denoising steps ($t = 2$ to $t = 49$).

Two key patterns emerge from these visualizations:

- **Modal Heterogeneity:** There is a distinct disconnect between the curvature landscapes of RGB and Depth modalities. For instance, in Prompt 2 (Step 37), the Depth tokens exhibit high curvature (indicated by purple regions) corresponding to large geometric structures, while the corresponding RGB tokens show a different, texture-dependent distribution. This confirms that a single caching decision cannot satisfy both modalities simultaneously.

- **Spatial Heterogeneity:** Within any single heatmap, the curvature is non-uniform. High-curvature "chaotic" regions (often object boundaries or rapid motion areas) coexist with extensive low-curvature "stable" regions (backgrounds). This spatial variance persists across different prompts, reinforcing the need for a spatially adaptive prediction mechanism.

## E.2. Effectiveness of Curvature-guided Grouping

We further validate our method through a quantitative ablation study, comparing WORLDCACHE against uniform prediction baselines and random grouping strategies. Table 5 in the main text reports the reconstruction quality (PSNR, SSIM, LPIPS) and generation latency.

**Failure of Uniform Strategies.**  Applying a single prediction rule to all tokens proves suboptimal regardless of the complexity of the operator:

- **Uniform Reuse:** While computationally efficient ($O(1)$), simply copying features leads to mediocre fidelity (PSNR 22.74). It fails to capture the evolution of features, particularly for tokens that are not in a steady state .

- **Uniform Linear:** Naively applying linear extrapolation to all tokens results in the worst performance (PSNR 18.01). This catastrophic drop is attributed to the presence of high-curvature "chaotic" tokens. In high-curvature regions, linear projection ignores the non-linear "bending" of the feature trajectory, causing severe overshooting and feature divergence that degrades the entire frame.

- **Uniform Damped:** While our adaptive damped prediction is designed to stabilize chaotic regions by interpolating between current and historical velocities, applying it uniformly to all tokens is also suboptimal. Although it outperforms the linear baseline by preventing divergence (PSNR 23.76 vs. 18.01), it yields a lower SSIM (0.665) compared to simple reuse (0.714). This suggests that for "stable" or "linear" tokens, the additional damping logic introduces unnecessary historical bias and smoothing, which can blur static details or lag behind predictable motions.

**Importance of Curvature-based Grouping.**  To isolate the contribution of our curvature metric, we compare our method against a **Random Grouping** baseline, which assigns tokens to Stable/Linear/Chaotic groups randomly (preserving the same ratios).

- **Random vs. Curvature:** Random grouping performs significantly worse than our method (e.g., SSIM 0.710 vs. 0.791). This confirms that the performance gain does not come merely from mixing different operators, but from correctly identifying *which* tokens require which operator.

- **CHTP Superiority:** Our proposed CHTP achieves the highest quality metrics (PSNR 25.76, LPIPS 0.227) with negligible latency overhead compared to the cheapest baseline. By accurately assigning reuse to stable tokens, linear extrapolation to smooth motion, and damped prediction to chaotic regions, CHTP effectively resolves the trade-off between stability and responsiveness.

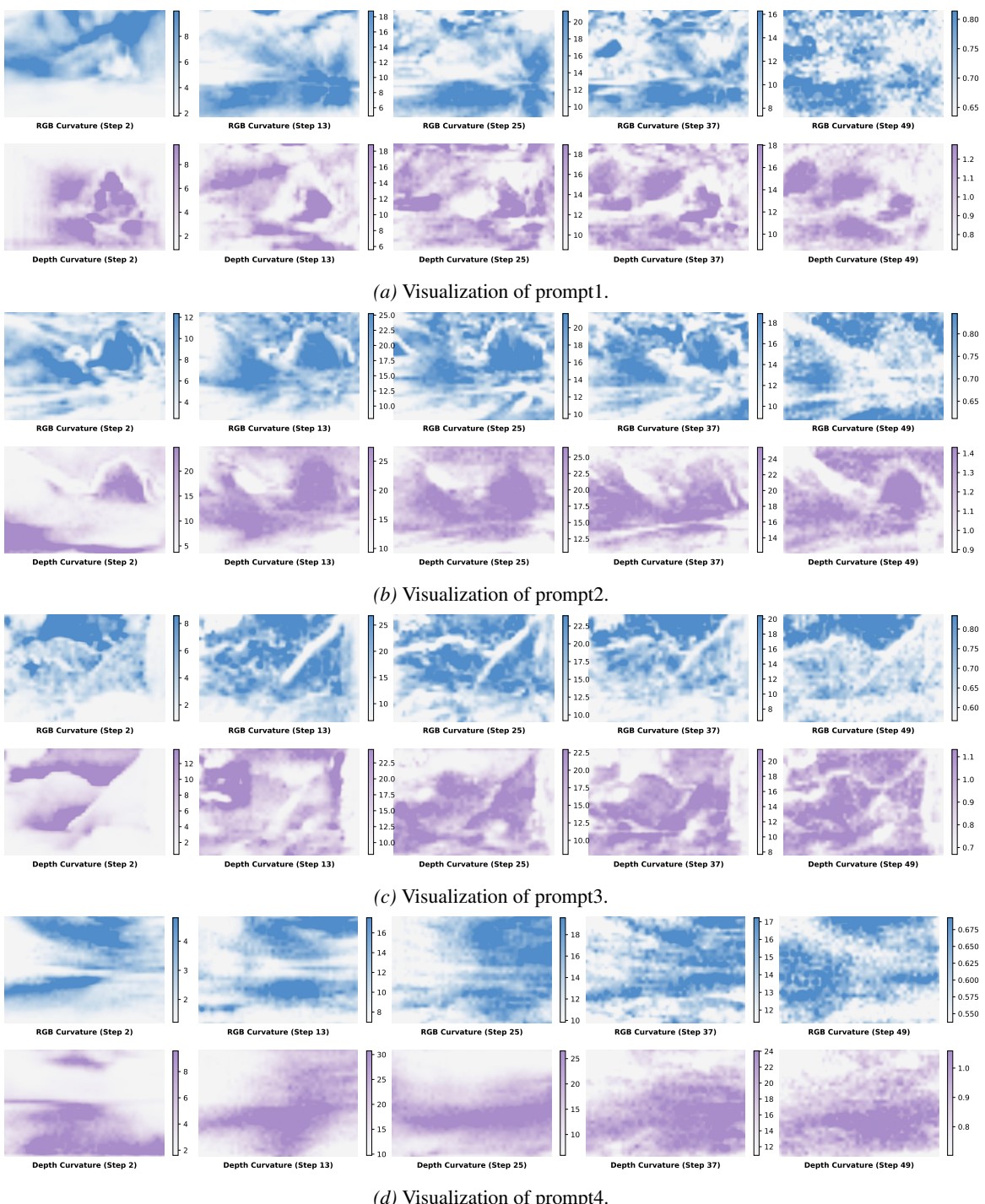

*(a)* Visualization of prompt1.

*(b)* Visualization of prompt2.

*(c)* Visualization of prompt3.

*(d)* Visualization of prompt4.

*Figure 7.* **More visualization of token heterogeneity.** We visualize both RGB tokens and Depth tokens of different prompts across 50 denoising steps in HunyuanVoyager (Huang et al., 2025). The distinct patterns between RGB and Depth, as well as the spatial variance within each map, underscore the necessity of heterogeneous processing.

## F. More Analysis of Chaotic-prioritized Adaptive Skipping

In this section, we provide a deeper investigation into the design rationale of our *Chaotic-prioritized Adaptive Skipping* (CAS) strategy. We present visual evidence of the non-uniform error distribution that necessitates a prioritized approach, illustrate the scale variance that mandates a dimensionless metric, and quantitatively benchmark CAS against alternative

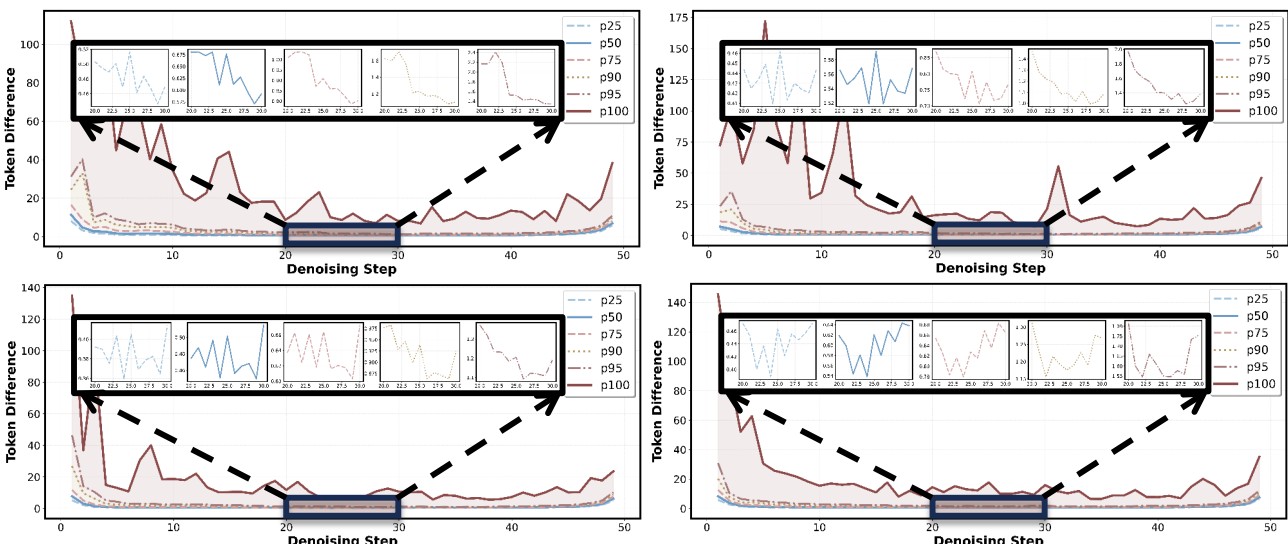

*Figure 8.* **Visualization of non-uniform temporal error dynamics.** We plot the prediction error accumulation across different token percentiles ($p_{25}$ to $p_{100}$). The results show that global variance is dominated by the top percentile of "chaotic" tokens (red line), while the majority of tokens ($p_{50}$ and below) remain stable. This supports our strategy of monitoring the *Chaotic* group rather than the global mean.

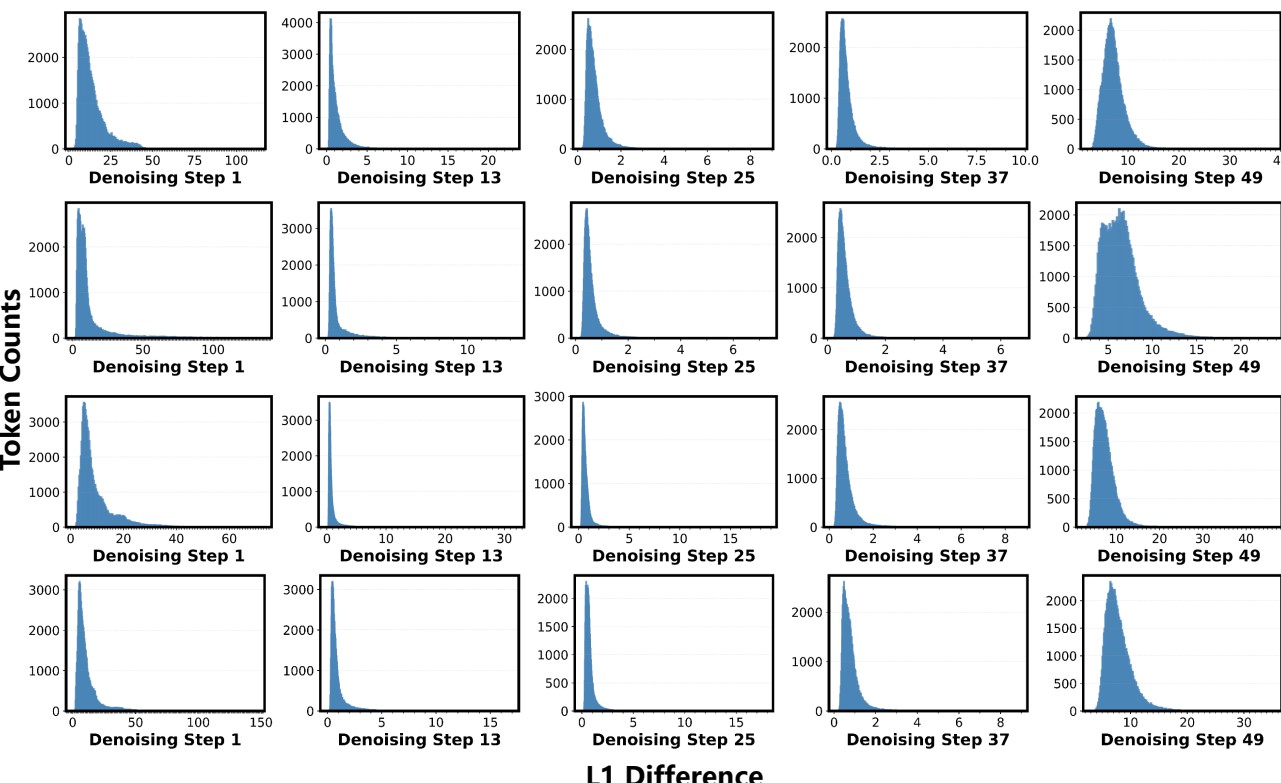

*Figure 9.* **Visualization of temporal scale variance.** Histograms show the distribution of feature values at different denoising steps. The numerical range fluctuates by orders of magnitude (e.g., thousands at Step 1 vs. single digits at Step 37), proving that fixed thresholds on raw values are unreliable and validating the necessity of our *dimensionless* drift indicator.

skipping heuristics.

### F.1. Dominance of Chaotic Tokens in Temporal Dynamics

A core premise of **WorldCache** is that global error metrics are diluted by the vast majority of stable tokens, masking the critical drift of "hard" tokens. To verify this, we visualize the temporal evolution of prediction errors (L1 difference between predicted and ground truth features) across different percentiles ($p_{25}$ to $p_{100}$) in Figure 8.

As observed in the plots, the error trajectories for lower percentiles (e.g., $p_{25}, p_{50}$, blue/grey lines) remain consistently flat and low throughout the denoising process. In contrast, the top percentile ($p_{100}$, red line), representing the most chaotic tokens, exhibits sharp, erratic spikes. Crucially, the global variance is almost entirely driven by this minority. A standard "average error" threshold would smooth out these spikes, failing to trigger a re-computation when the chaotic tokens diverge. By explicitly tracking the *Chaotic* group, our method captures these critical failure points that standard metrics miss.

### F.2. Necessity of Dimensionless Indicators

Another challenge in dynamic skipping is setting a robust threshold. As shown in Figure 9, the distribution of feature magnitudes and error scales varies drastically across different denoising timesteps. For example, early steps (e.g., Step 1) may exhibit error scales in the hundreds (0-100), while later steps (e.g., Step 37) cluster in a much narrower, smaller range (0-10).

This scale variance renders 'raw' metrics (like absolute L1 distance) ineffective: a threshold $\tau$ suitable for Step 37 would trigger at every single iteration in Step 1, while a threshold suitable for Step 1 would never trigger in Step 37. This empirical evidence justifies our design of the **Dimensionless Drift Indicator** ($E = \kappa \cdot \|\Delta y\|$). By normalizing the displacement with curvature, we obtain a scale-invariant metric that robustly indicates relative instability across all phases of the denoising process.

### F.3. Ablation on Skipping Strategies

Finally, we quantitatively compare our CAS strategy against other common skipping policies in Table 6 of the main text.

- **Fixed Interval:** A rigid "skip-$k$-compute-1" schedule yields the baseline performance (PSNR 26.18). It fails to adapt to the variable difficulty of steps.

- **Difference/Norm Guided:** Using raw Feature Difference $\left\|\mathbf{y}_t - \mathbf{y}_{t+1}\right\|_2$ or Norm $\frac{\left\|\mathbf{y}_t - \mathbf{y}_{t+1}\right\|_2}{\left\|\mathbf{y}_{t+1}\right\|_2}$ as triggers improves over fixed intervals (PSNR 26.79) but is suboptimal due to the scale variance issue discussed above, where the thresholds are hard to tune globally.

- **Curvature Guided:** Triggering based solely on curvature (difficulty) without considering actual displacement (movement) performs poorly (PSNR 25.87). High curvature implies a *potential* for error, but if the token displacement is small, re-computation is wasteful.

- **CAS (Ours):** Our method achieves the best trade-off (PSNR 27.10, SSIM 0.881). By combining curvature (potential difficulty) with displacement (actual drift) into a dimensionless metric, and prioritizing the chaotic group, CAS ensures resources are allocated exactly when and where the model is most likely to fail.

## G. Reproducibility Statement

To enhance reproducibility, we have attached our necessary code and the generated raw video files in the supplementary material.

## H. More Visual Comparison

Here, we provide more visual comparisons to demonstrate the effectiveness of our proposed **WorldCache**. The results are shown in Fig. 10, Fig. 11, Fig. 12, and Fig. 13.

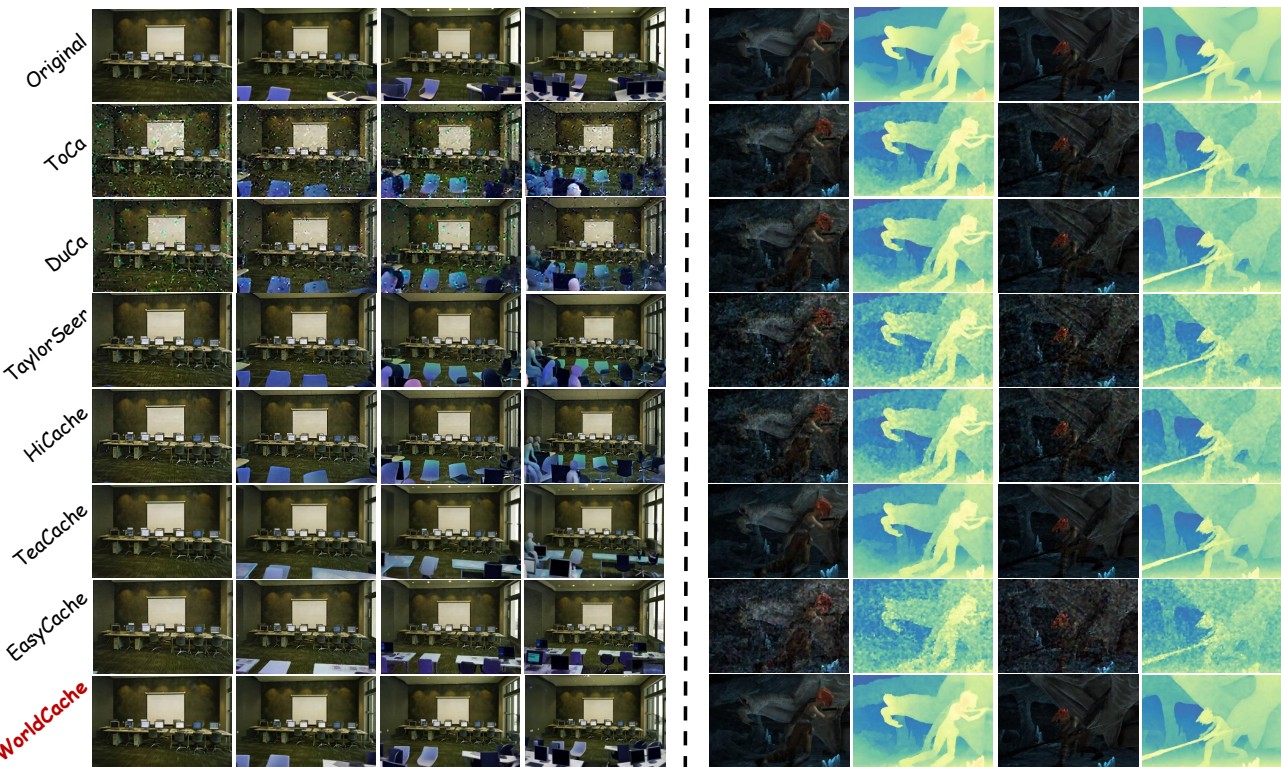

*Figure 10.* More visual comparison between **WorldCache** and existing methods.

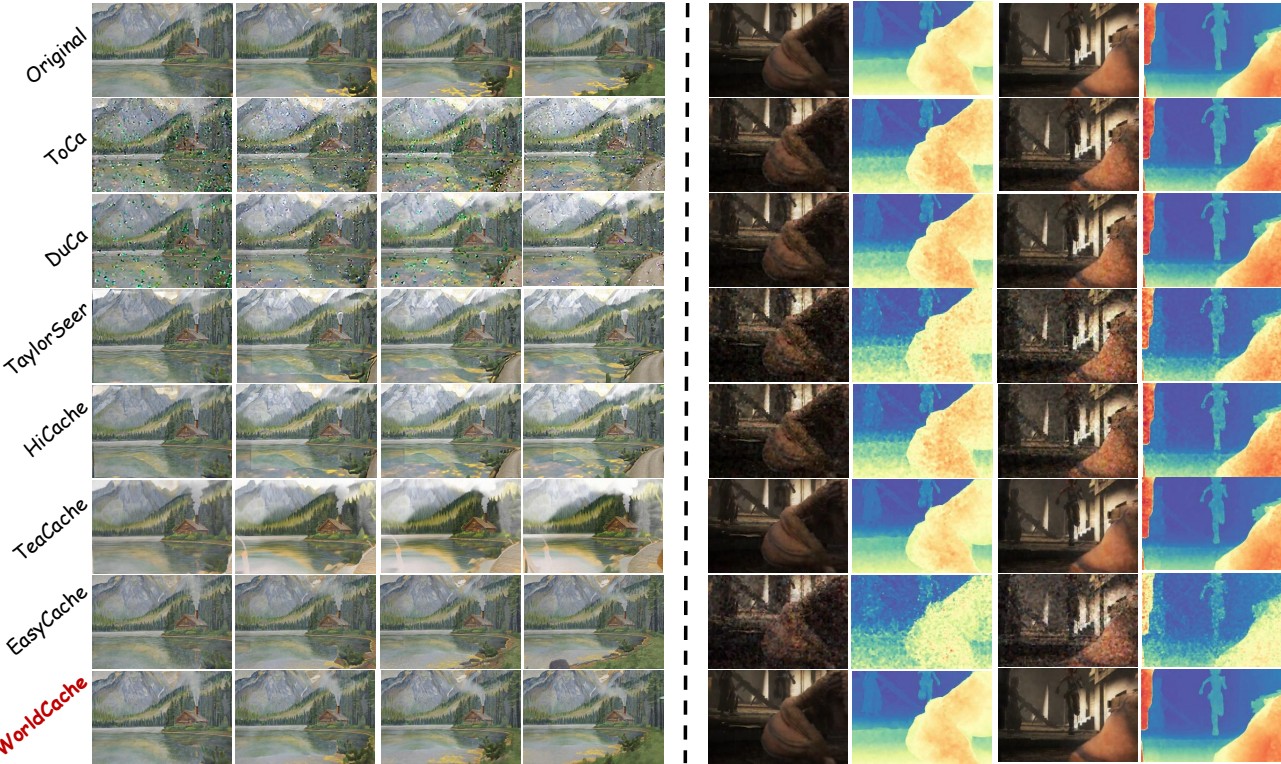

*Figure 11.* More visual comparison between **WorldCache** and existing methods.

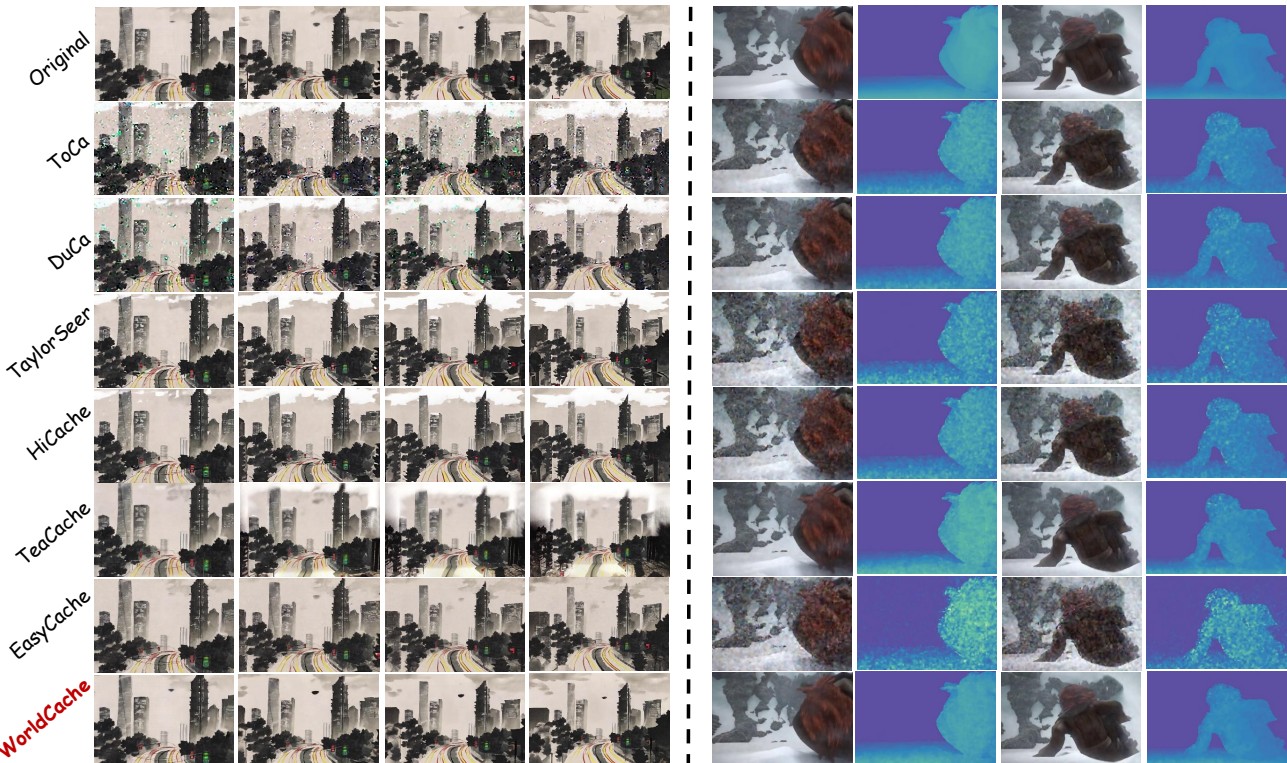

*Figure 12.* More visual comparison between **WorldCache** and existing methods.

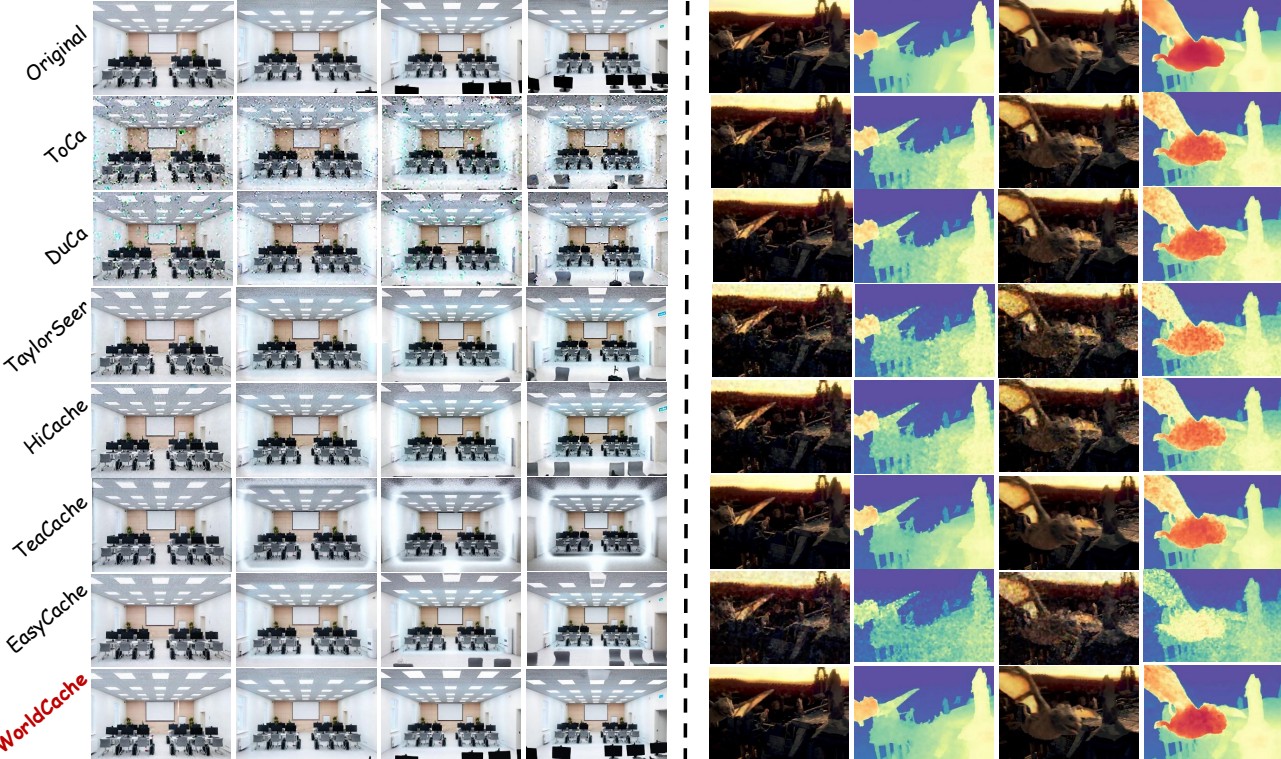

*Figure 13.* More visual comparison between **WorldCache** and existing methods.

