# OpenReview forum: "WorldCache: Accelerating World Models for Free via Heterogeneous Token Caching"
_ICML.cc/2026/Conference — ICML 2026 regular_

### Official Review · Reviewer_NsUS · 2026-03-08

**Soundness:** 3
**Presentation:** 3
**Significance:** 2
**Originality:** 3
**Overall Recommendation:** 5
**Confidence:** 3

**Summary:**

This paper studies the problem of improving inference efficiency in diffusion-based world models. Recent work has explored the use of diffusion models, to serve as world models capable of generating high-fidelity environment rollouts conditioned on the current state and actions. While these models show strong generative capabilities, their practical deployment is hindered by the high computational cost of diffusion sampling, which requires many iterative denoising steps and makes long-horizon simulation expensive. To address this issue, the authors propose WorldCache, a training-free acceleration framework designed specifically for diffusion-based world models. The key idea is that tokens within world model representations exhibit heterogeneous temporal dynamics during diffusion sampling: some tokens evolve smoothly and are easy to predict, while others exhibit chaotic behavior and dominate prediction errors. Based on this observation, the method introduces two components: a curvature-guided token prediction mechanism that estimates the predictability of each token and applies different prediction strategies accordingly, and a chaotic-prioritized adaptive skipping strategy that selectively recomputes tokens whose predicted errors exceed a threshold. Therefore, these mechanisms aim to reduce the number of expensive model evaluations during diffusion sampling while preserving rollout quality.

**Compliance With Llm Reviewing Policy:**

Affirmed.

**Final Justification:**

After careful consideration of the paper’s strengths, weaknesses, and overall contribution, I confirm that I will keep my final score as this.

**Key Questions For Authors:**

1. The method is evaluated on two diffusion-based world models. How well does the approach generalize to other diffusion-based world model architectures or simulation settings beyond those tested in the paper?

2. The proposed caching mechanism introduces a trade-off between computational efficiency and prediction accuracy. Could the authors discuss more clearly the limitations of this trade-off? In particular, under what conditions might the approximation lead to noticeable degradation in rollout quality?

**Limitations:**

Yes.

**Strengths And Weaknesses:**

Strength:
1. The paper addresses an important practical problem. Diffusion-based world models require many denoising steps during inference, which makes long-horizon rollout computationally expensive. Improving inference efficiency is therefore important for scaling world models.
2. The method is motivated by a reasonable empirical observation: tokens in diffusion world models show heterogeneous temporal dynamics. Using a curvature-based metric to estimate token predictability and decide when to reuse cached features is a sensible heuristic.
3. The proposed framework is well-presented. The two main components (curvature-guided token prediction and chaotic-prioritized skipping) are clearly motivated and follow an intuitive design.

Weakness:
1. The overall idea is closely related to prior work on diffusion inference acceleration and token caching (e.g., TeaCache, ToCa, EasyCache). The main contribution appears to be adapting existing ideas to world models rather than introducing fundamentally new mechanisms.
2. It is unclear whether the proposed method truly relies on properties specific to world models. Many components of the approach seem applicable to general diffusion models, which weakens the claim that the design is uniquely motivated by world model dynamics.

---

> ### Author Rebuttal · Authors · 2026-03-30
>
> We greatly appreciate your positive feedback on our work. Below we clarify your concerns in more depth.
>
> > **W1 & W2: About core contribution / novelty and properties specific to world models.**
>
> We thank the reviewer for this important comment. Our claim is **not** that caching, token-adaptive processing, or drift-triggered FULL recomputation are individually new. Rather, our novelty is to identify and formalize a **world-model-specific failure regime** for diffusion caching, and then design the cache mechanism around it. In other words, our contribution is not a new primitive in isolation, but a diagnosis of **why directly transferring existing diffusion caching policies becomes unreliable in world models**. Many prior caching methods work reasonably well when feature dynamics are relatively homogeneous, but this assumption breaks in diffusion world models. Concretely, the difference from prior methods is threefold.
>
> 1. **Uniform cache/update rules are insufficient.** Existing methods often apply one approximation rule to all tokens. In world models, however, **long-tailed token heterogeneity** arises from multi-modal coupling and strong spatial variation, so token difficulty is highly uneven across modalities and regions. A single rule is therefore either too conservative for easy tokens or unstable on the hard minority. This is why WorldCache uses **curvature-based three-way grouping with group-specific predictors**, rather than one shared rule for all tokens.
>
> 2. **Global or token-agnostic triggers are insufficient.** In world models, failure is often governed by a small **bottleneck subset** of hard tokens, while most tokens remain stable. As a result, global error statistics can be **diluted by easy tokens** and react too late to the true failure points. This is why our **chaotic-prioritized monitoring** tracks the hardest subset and triggers FULL computation only when that subset begins to drift.
>
> 3. **Raw, scale-dependent triggers are insufficient.** Raw feature differences or norm-based thresholds vary substantially across modalities and denoising steps, making a unified threshold unreliable in the multi-modal world-model regime. A threshold that is reasonable at one step may over-trigger early and under-trigger late. This is why CAS uses a **dimensionless curvature-normalized drift** score rather than a raw trigger.
>
> To make the first point concrete, we compare token heterogeneity between HunyuanVideo-13B (a pure video diffusion model) and Voyager-13B (world model) using curvature variance:
>
> Model|Curvature Variance
> -|-
> HunyuanVideo-13B|13.52
> Voyager-13B|37.76
>
> This gap shows that token heterogeneity is much stronger in the world-model setting, consistent with Fig. 3 and Fig. 7. It explains why a **single cache/update rule is ill-matched** to world models, and why the true bottleneck is not the average token but the hard, heterogeneous minority.
>
> The practical benefit is reflected in Table 1: on Voyager-13B, WorldCache improves over **EasyCache** from **21.76 to 23.49 PSNR**, achieves slightly higher speedup (**3.65x vs. 3.58x**), and keeps memory essentially unchanged (**50.58 GB vs. 50.44 GB baseline**). Thus, the gain does not come from a heavier mechanism, but from matching the policy to the actual failure structure of world models.
>
> Therefore, our claim is not “a universally new caching primitive,” but a **world-model-motivated reformulation of why prior caching policies fail, together with a matched mechanism for heterogeneous token regimes**. While some components may transfer more broadly, the **diagnosis, reformulation, and mechanism composition** are driven by the multi-modal, spatially varying, and hard-token-dominated dynamics that are particularly severe in world models.
>
> >**Q1: Evaluation on broader settings.**
>
> We conduct more evaluations, including additional world model LingBot-14B, multiple resolutions, more denoising steps, and longer rollouts. Please see our rebuttal to **Reviewer iAXy `W2 and Q1: Evaluation on more settings`**.
>
> >**Q2: Trade-off between efficiency and accuracy.**
>
> We agree that feature caching inevitably introduces this trade-off. To make this explicit, we now include the trade-off results in Table 4 by varying the adaptive-skipping threshold $\eta$:
>
> Method|PSNR$\uparrow$|SSIM$\uparrow$|Latency (s)$\downarrow$
> -|-|-|-
> TeaCache|26.60|0.843|114.2
> CAS (η=0.10)|31.66|0.922|109.0
> CAS (η=0.15)|31.13|0.916|108.5
> CAS (η=0.20)|30.63|0.908|107.2
> CAS (η=0.25)|29.22|0.907|99.35
> CAS (η=0.30)|28.05|0.894|93.86
> CAS (η=0.35)|27.10|0.881|90.35
>
> As shown in Table 4, increasing $\eta$ yields a smooth transition from **higher fidelity** to **lower latency**, confirming that CAS provides a controllable accuracy–efficiency trade-off. Importantly, even at the aggressive setting $\eta=0.25$, CAS is still both way faster and better on all three perceptual metrics than TeaCache.
>
> For failure analysis, please see our rebuttal to **Reviewer iAXy, `Q3: Failure analysis`**.

---

> > ### Author Rebuttal · Reviewer_NsUS · 2026-04-01
> >
> > My concerns have been adequately addressed.

---

> > > ### Author Response · Authors · 2026-04-02
> > >
> > > Dear reviewer,
> > >
> > > We sincerely appreciate your time and constructive feedback throughout the review process. We are delighted to hear that our rebuttal has addressed your concerns and that you now **recommend clear acceptance**. Your insightful comments have significantly strengthened our paper, and we are grateful for your valuable contribution to improving our work.
> > >
> > > Best wishes,
> > >
> > > All authors

---

### Official Review · Reviewer_anti · 2026-03-10

**Soundness:** 4
**Presentation:** 4
**Significance:** 4
**Originality:** 4
**Overall Recommendation:** 4
**Confidence:** 4

**Summary:**

The paper introduces worldcache, a framework designed to accelerate diffusion world models. Diffusion world models are generative systems that simulate environments and physical dynamics by iteratively removing noise from multi-modal latents. Feature caching, a technique that reuses past computations to skip expensive network evaluations, struggles in this domain due to token heterogeneity and non-stationary temporal dynamics. Worldcache addresses this by introducing curvature-guided heterogeneous token prediction, which calculates a curvature score to measure how predictably a token's feature trajectory bends over time. Tokens are partitioned into stable, linear, and chaotic groups. Stable tokens are directly reused, linear tokens are extrapolated, and chaotic tokens use a damped update guided by a hermite polynomial, which is a smooth, s-shaped mathematical curve used to blend values without overshooting.  The framework also employs chaotic-prioritized adaptive skipping, which monitors a dimensionless drift score exclusively on the chaotic tokens to safely determine when a full model evaluation is necessary. Experiments demonstrate up to 3.7x speedups with minimal quality loss.

**Compliance With Llm Reviewing Policy:**

Affirmed.

**Final Justification:**

The rebuttal and additional experiments have addressed the technical concerns regarding hyperparameter sensitivity, memory scalability, and block-level applicability. The supplementary evidence supports the claims of the curvature-guided partitioning framework.

**Key Questions For Authors:**

How sensitive is the framework to the choice of the small constant epsilon in the curvature calculation?

Could the curvature-guided token partitioning be applied dynamically at the architectural block level rather than just at the model-output level?

How does the memory overhead scale when increasing the context length or frame count beyond the tested 49 frames?

**Limitations:**

Yes, the authors provided a brief impact statement. It would be beneficial to add a discussion on specific failure modes, such as whether highly dynamic scenes with rapid camera movements might cause the chaotic token set to grow too large, thus negating the acceleration benefits.

**Strengths And Weaknesses:**

Soundness: the submission is technically rigorous. The physical motivation for using curvature to assess predictability is well-founded, and the proof for the dimensionless normalization demonstrates why raw feature differences fail across timesteps. The empirical results strongly support the claims. A minor weakness is the reliance on fixed quantiles for token grouping, which might require tuning for entirely new datasets.
Presentation: the writing is clear and the narrative is logically structured. The visualizations effectively illustrate the spatial and modality-based variance of tokens. Significance: reducing the inference cost of simulating environments is a pressing need for interactive agents. The insights regarding long-tailed difficulty and non-uniform temporal dynamics in multi-modal generation are highly relevant and could influence future acceleration methods.
Originality: applying token-adaptive caching based on geometric trajectory curvature, specifically tailored for the chaotic nature of world models, is a fresh and creative perspective.

---

> ### Author Rebuttal · Authors · 2026-03-30
>
> Thank you for the careful review and your helpful suggestions. We respond to your questions as follows.
>
> >**W1: The reliance on fixed quantiles for token grouping.**
>
> Empirically, the method is **not highly sensitive** to the exact percentile choices. We already studied this in **Table 3**, where we vary $p_s$ and $p_c$ over a broad range. The results show that percentile-based grouping is **not brittle**, and fixed quantiles serve as a **practical default** rather than delicately tuned dataset-specific parameters. We reproduce the ablation below:
>
> |$p_s,p_c$|PSNR$\uparrow$|SSIM$\uparrow$|LPIPS$\downarrow$|
> |-|-|-|-|
> EasyCache|21.76|0.737|0.208
> {0.3,0.8}|23.32|0.766|0.179
> {0.3, 0.7}|23.49|**0.770**|**0.176**
> {0.3, 0.6}|**23.52**|0.768|0.178
> {0.2, 0.8}|23.12|0.760|0.185
> {0.2, 0.7}|23.12|0.764|0.182
> {0.2, 0.6}|23.33|0.764|0.181
> {0.1, 0.8}|22.77|0.758|0.188
> {0.1, 0.7}|22.97|0.758|0.185
> {0.1, 0.6}|22.95|0.758|0.187
>
> First, there is a clear **robustness plateau** around the default setting. Across all 9 tested pairs, PSNR stays within **22.77-23.52**, SSIM within **0.758-0.770**, indicating that the grouping design is robust rather than fragile.
>
> Second, the improvement is **not tuning-driven**. All 9 tested settings outperform the strongest uniform baseline EasyCache. Even the weakest pair, $(0.1,0.8)$, still improves over EasyCache by **+1.01 PSNR** and **+0.021 SSIM**, while the best setting brings up to **+1.76 PSNR** and **+0.033 SSIM**. This suggests that the main gain comes from the heterogeneous-grouping principle itself, not from optimizing one narrow percentile pair.
>
> Therefore, fixed quantiles should be viewed as a **simple and practical default** for separating stable / linear / chaotic regimes, rather than a fragile parameter choice. On a new dataset, the best pair may shift slightly, but Table 3 indicates a **broad good region** rather than a narrow optimum.
>
> >**Q1: Sensitivity of small constant epsilon in the curvature calculation.**
>
> The small constant $\varepsilon$ is introduced only as a standard numerical-stability term to avoid division by zero when $|v_i|_2^2$ becomes extremely small; it is **not a tuning parameter** of the method.
>
> To verify this, we conducted an ablation with three values of $\varepsilon$:
>
> |$\varepsilon$|PSNR$\uparrow$|SSIM$\uparrow$|LPIPS$\downarrow$|
> |-|-|-|-|
> $1e^{-10}$|23.51|0.77|0.175
> $1e^{-9}$|23.49|0.77|0.176
> $1e^{-8}$|23.49|0.77|0.176
>
> The results are nearly identical across all settings, indicating that the framework is **insensitive** to the choice of $\varepsilon$ within a reasonable range.
>
> >**Q2: Applying curvature-guided token partitioning in block-level.**
>
> Yes. We implemented a block-level variant of WorldCache on Aether-5B, and it remains effective. This suggests that our curvature-guided partitioning is not tied to model-output caching; however, the model-level version remains preferable for our target low-memory world-model setting.
>
> |Method|PSNR$\uparrow$|SSIM$\uparrow$|LPIPS$\downarrow$|Memory (GB)$\downarrow$|
> |-|-|-|-|-|
> HiCache|24.93|0.784|0.226|77.32
> HERO|23.56|0.741|0.259|75.08|
> **WorldCache (block-level)**|29.72|0.898|0.092|62.02
> **WorldCache (model-level)**|**31.87**|**0.924**|**0.066**|**46.59**
>
> Compared with the strongest block-level baseline (HiCache), **WorldCache (block-level)** substantially improves quality while also reducing memory usage (**+4.79 PSNR** and **15.30 GB** lower memory). At the same time, the **model-level** version performs even better, which is why we adopt it as the main design.
>
> >**Q3: memory overhead scale when increasing the context length.**
>
> In our tested range, the extra memory of **WorldCache remains negligible as context length increases**. The reason is that WorldCache is a **model-level** caching method: it stores only a **fixed-depth output-level history** (the last 3 FULL outputs) and token masks, rather than per-layer activations. Thus, the dominant memory growth still comes from the backbone processing a longer token sequence, which is shared by both the baseline and WorldCache.
>
> We verify this with an ablation on LingBot-14B:
>
> Context Length|Baseline Mem. (GB)$\downarrow$| WorldCache Mem. (GB)$\downarrow$|Extra Mem. (GB)$\downarrow$|Speedup$\uparrow$|
> -|-|-|-|-
> 49|49.37|49.39|0.02|2.41x
> 81|55.87|55.89|0.02|2.47x
> 161|71.32|71.35|0.03|2.53x
>
> As the context grows from **49 to 161 frames** $(3.3\times)$, the baseline peak memory increases by **21.95 GB**, whereas the additional memory introduced by WorldCache changes only from **0.02 GB** to **0.03 GB** (below **0.05%** of baseline memory). Meanwhile, the speedup remains stable or slightly improves (2.41x → 2.53x).
>
> Therefore, in this tested range, **WorldCache preserves the low-memory advantage of model-level caching, and its incremental memory does not grow materially with longer contexts**.
>
> >**Limitation: About failure analysis.**
>
> For failure analysis, please see our rebuttal to **Reviewer iAXy, `Q3: Failure analysis`**.

---

> > ### Author Rebuttal · Reviewer_anti · 2026-04-03
> >
> > My concerns have been adequately addressed. Thanks for the detailed rebuttal.

---

> > > ### Author Response · Authors · 2026-04-06
> > >
> > > Dear reviewer,
> > >
> > > We sincerely appreciate your time and constructive feedback throughout the review process. We are delighted to hear that **your concerns have been addressed**. We are grateful for your careful evaluation and valuable support of our work.
> > >
> > > Best wishes,
> > >
> > > All authors

---

### Official Review · Reviewer_iAXy · 2026-03-13

**Soundness:** 3
**Presentation:** 3
**Significance:** 3
**Originality:** 3
**Overall Recommendation:** 4
**Confidence:** 3

**Summary:**

This paper proposes WorldCache, a training-free acceleration method for diffusion world models. The key observation is that token dynamics are highly heterogeneous across space and modality, and inference failures are often driven by a small subset of difficult tokens rather than by average drift. The method combines curvature-guided heterogeneous token prediction and chaotic-prioritized adaptive skipping, which triggers full recomputation based on an accumulated drift score over hard tokens. Experiments on Voyager-13B and Aether-5B show clear speed-quality improvements over prior caching baselines, and additional 3D reconstruction results suggest that the method largely preserves geometry-related capability.

**Compliance With Llm Reviewing Policy:**

Affirmed.

**Key Questions For Authors:**

1.	How robust is the proposed skipping criterion across different samplers, denoising step counts, and rollout lengths?
2.	Can the authors better clarify the scope of the theoretical claim around the curvature-normalized drift score?
3.	What are the dominant failure?
4.	What fraction of end-to-end runtime is spent on the control logic itself?
5.	How sensitive is performance to the percentile-based grouping design?

**Limitations:**

The paper should discuss its limited model coverage and provide a clearer account of failure modes and hyperparameter robustness.

**Strengths And Weaknesses:**

Strengths:

1.	The paper innovatively identifies that world-model acceleration is harder than standard diffusion acceleration because token predictability is highly non-uniform, and error growth is dominated by a small hard subset.
2.	The proposed design is simple, well aligned with this observation, and effective in practice.
3.	The experimental results are strong on the tested models, and the ablations are generally convincing.

Weaknesses:

1.	My main concern is that the empirical evidence is stronger than the theoretical justification. The curvature-based design is intuitive and appears effective, but the paper’s theory—especially the physics-grounded motivation and the scale-normalized drift argument—does not fully support the scope of the claims. In my view, this is better understood as a strong and well-motivated heuristic than as a principled theoretical contribution.
2.	My second concern is the evaluation scope. The results on Voyager and Aether are promising, but they do not yet demonstrate robustness across a wider range of world models, samplers, denoising schedules, or longer rollouts. The paper would also benefit from a clearer failure analysis, such as identifying the types of motion or geometry changes where caching begins to fail. Finally, for a deployment-focused paper, it would be useful to report the runtime overhead of curvature estimation and control logic relative to backbone computation.

---

> ### Author Rebuttal · Authors · 2026-03-30
>
> We sincerely thank you for your positive assessment of our work and your constructive suggestions. We address your comments in detail below.
>
> >**W1 and Q2: Theoretical justification for curvature choice and scope of Theorem 4.1.**
>
> For the theoretical justification for curvature choice and scope of Theorem 4.1, please see our rebuttal to **Reviewer TE6u `W2: Theoretical justification for curvature choice.`**.
>
>
> >**W2 and Q1: Evaluation on more settings.**
>
> Thank you for your advice. To address this concern, we add new experiments along four additional axes beyond the original Voyager/Aether results: **(i) a third diffusion world model (LingBot-14B [1]), (ii) multi-resolution evaluation, (iii) different denoising schedules, and (iv) longer rollouts**; across all new settings, WorldCache continues to deliver the best quality-efficiency trade-off.
>
> **1) New model + multi-resolution (LingBot-14B, 49f, 70 steps).** The advantage persists on a new backbone and at both resolutions.
>
> |Method|Resolution|PSNR$\uparrow$|SSIM$\uparrow$|Latency (s)$\downarrow$|
> |-|-|-|-|-
> HiCache|464×832|17.82|0.678|451.6 (1.86x)
> EasyCache|464×832|20.28|0.751|388.1 (2.16x)
> **WorldCache**|464×832|**23.24**|**0.812**|**348.7 (2.41x)**
> HiCache|720×1280|18.39|0.688|2820 (0.95x)
> EasyCache|720×1280|20.93|0.762|1189 (2.25x)
> **WorldCache**|720×1280|**23.99**|**0.824**|**1066 (2.51x)**
>
> **2) Different denoising schedules (Voyager-13B, 49f, 512x768).** The benefit also holds under both 70-step and 30-step sampling.
>
> |Method|Denoise Steps|PSNR$\uparrow$|SSIM$\uparrow$|Latency (s)$\downarrow$|
> |-|-|-|-|-
> HiCache|70|20.06|0.667|1348 (1.07x)
> EasyCache|70|23.56|0.789|356.3 (4.03x)
> **WorldCache**|70|**25.49**|**0.828**|**351.2 (4.09x)**
> HiCache|30|18.03|0.616|852.1 (0.71x)
> EasyCache|30|21.12|0.728|232.7 (2.61x)
> **WorldCache**|30|**22.78**|**0.760**|**223.3 (2.72x)**
>
> **3) Longer rollouts (LingBot-14B, 464x832, 70 steps).** WorldCache remains state-of-the-art to 161 frames.
>
> |Method|Frames|PSNR$\uparrow$|SSIM$\uparrow$|Latency (s)$\downarrow$
> |-|-|-|-|-
> EasyCache|49|20.28|0.751|388.1 (2.16x)
> **WorldCache**|49|**23.24**|**0.794**|**348.7** (**2.41x**)
> EasyCache|81|21.81|0.705|677.4 (2.29x)
> **WorldCache**|81|**23.37**|**0.780**|**626.5** (**2.47x**)
> EasyCache|161|21.81|0.777|1679.3 (2.45x)
> **WorldCache**|161|**22.16**|**0.803**|**1626.0** (**2.53x**)
>
> WorldCache is now validated on **three diffusion world models** across multiple **resolutions, denoising schedules, and rollout lengths**, and it consistently achieves a better quality-speed trade-off in all tested settings.
>
> [1]. Advancing Open-source World Models. (arxiv 2026)
>
> >**Q3: Failure analysis.**
>
> Regarding limitations, degradation is most likely in highly dynamic regimes, especially under rapid camera motion and complex textures. In these cases, failure is mostly driven by a small chaotic-token subset whose spikes dominate the error. Thus aggressive caching may trigger more FULL recomputation, shrinking speedup, or cause visible drift; our CAS is designed to mitigate this, but does not completely eliminate this trade-off.
>
> We will revise the paper to make this clearer.
>
> >**Q4: Runtime overhead.**
>
> We now explicitly measured the runtime of **curvature estimation + grouping + control logic**. Across three models, this cost is only **0.064-0.189s**, i.e., **0.047%-0.060%** of end-to-end latency (about **0.05%** overall):
>
> Model|WorldCache Overhead (s)|End-to-End Latency (s)|Overhead / Latency
> -|-|-|-
> LingBot-14B|0.189|348.7|0.054%
> Voyager-13B|0.137|288.6|0.047%
> Aether-5B|0.064|107.2|0.060%
>
> All end-to-end latencies reported in the paper already include this control path. Consistently, Table 5 shows that **CHTP** remains latency-comparable to simpler predictors (**86.94s** vs. **86.32-87.51s**) while achieving the best quality; in particular, it is faster than uniform **Linear** (**87.07s**) and **Damped** (**87.51s**). Thus, the control overhead itself is negligible at about **0.05%**, and the observed end-to-end speedups (**2.41x / 3.65x / 1.68x**) come from cheaper prediction on easy groups rather than being offset by the control logic.
>
>
> >**Q5: Sensitivity of percentile-based grouping design.**
>
> We have conducted an ablation study on percentile-based grouping design and show that our design is not sensitive to the parameter choice. Please see our rebuttal to **Reviewer anti `W1: The reliance on fixed quantiles for token grouping`**.

---

> > ### Author Rebuttal · Reviewer_iAXy · 2026-04-04
> >
> > Thank you for the detailed response. The rebuttal addresses most of my concerns and strengthens the paper. The additional experiments on a third world model, multiple resolutions, different denoising schedules, and longer rollouts make the empirical case substantially stronger. The added clarification on runtime overhead and failure cases is also useful.
> >
> > I still think the theoretical justification is less strong than the empirical evidence, but the rebuttal makes the intended role of the curvature-based design clearer. Overall, my main concerns are largely addressed, with some remaining reservations on the theoretical framing.

---

> > > ### Author Response · Authors · 2026-04-05
> > >
> > > Thank you for this follow-up. We here provide a more precise theoretical framing:
> > >
> > > Our intended claim is the following: in diffusion world models, the relevant property for feature caching is not raw motion magnitude, but token-wise **prediction hardness** under strong heterogeneity. A skipped token update is approximated by a cheap first-order surrogate, such as direct reuse or linear extrapolation. Therefore, cache difficulty should be defined by the approximation error of this first-order surrogate, rather than by first-order quantities such as feature difference or norm.
> > >
> > > To make this point precise, we will revise the paper around the following local statements.
> > >
> > > **Lemma 1. First-order cache error is governed by second-order variation.**
> > >
> > > Let $y_i(t)$ denote the feature trajectory of token $i$, and assume $y_i$ is $C^2$ on $[t, t+\Delta]$. Consider the first-order cache surrogate $y_i(t)+\Delta y_i’(t)$. Then the skipped-step cache error satisfies
> > >
> > > $$
> > > \left| y_i(t+\Delta)-\bigl(y_i(t)+\Delta y_i’(t)\bigr) \right|
> > > \le
> > > \frac{\Delta^2}{2}\sup_{\xi\in[t,t+\Delta]}|y_i’’(\xi)|.
> > > $$
> > >
> > > Thus, for first-order caching, the relevant notion of difficulty is not simply how far a token moves, but how much its trajectory departs from local linearity. The leading cache error is governed by second-order local dynamics.
> > >
> > > **Lemma 2. Any criterion based only on first-order difference cannot identify cache difficulty in general.**
> > >
> > > Consider two local trajectories:
> > > $$
> > > y_1(t+\tau)=p+u\tau,
> > > \qquad
> > > y_2(t+\tau)=p+u\tau+\frac{1}{2}a\tau^2,
> > > $$
> > > where $y_1$ is a pure linear one and $y_2$ has second-order item. At time $t$, they share the same position and first-order local behavior:
> > > $$
> > > y_1(t)=y_2(t)=p,
> > > \qquad
> > > y_1’(t)=y_2’(t)=u.
> > > $$
> > >
> > > However, their first-order cache errors over a skipped interval $\Delta$ are
> > > $$
> > > e_1(\Delta)=
> > > \left|y_1(t+\Delta)-\bigl(y_1(t)+\Delta y_1’(t)\bigr)\right|=0,
> > > $$
> > > while
> > > $$
> > > e_2(\Delta)=
> > > \left|y_2(t+\Delta)-\bigl(y_2(t)+\Delta y_2’(t)\bigr)\right|
> > > =\frac{1}{2}|a|\Delta^2.
> > > $$
> > >
> > > Therefore, two tokens can have the same first-order motion but very different cache difficulty. A complementary counterexample is a purely linear trajectory $y(t+\tau)=p+u\tau$, for which
> > >
> > > $$
> > > |y(t+\Delta)-y(t)|=|u|\Delta,
> > > $$
> > > can be arbitrarily large, while the first-order cache error is still exactly zero. Hence, raw difference measures displacement magnitude, not predictability under a first-order cache surrogate.
> > >
> > > **Lemma 3. Curvature provides a local upper bound on first-order cache difficulty.**
> > >
> > > Curvature is first motivated by its physical meaning as the local turning rate of a token trajectory. Define the smooth-limit curvature quantity
> > > $$
> > > \kappa_i(\xi)=\frac{|y_i’’(\xi)|}{|y_i’(\xi)|_2^2}.
> > > $$
> > >
> > > Given the token speed is locally bounded on $[t,t+\Delta]$ by
> > > $$
> > > |y_i’(\xi)| \le v_{\max},
> > > $$
> > > then
> > > $$
> > > |y_i’’(\xi)| \le v_{\max}^2 \kappa_i(\xi),
> > > $$
> > > and therefore
> > > $$
> > > \left| y_i(t+\Delta)-\bigl(y_i(t)+\Delta y_i’(t)\bigr) \right|
> > > \le
> > > \frac{\Delta^2}{2}v_{\max}^2\sup_{\xi\in[t,t+\Delta]}\kappa_i(\xi).
> > > $$
> > >
> > > Therefore, under a local speed bound, **curvature directly upper-bounds first-order cache error**. This is the precise sense in which curvature is better aligned than raw difference with cache difficulty: cache error is fundamentally a second-order effect, while raw difference is only a first-order quantity and can be large even when the true cache error is zero.
> > >
> > > **Consequence for our method.** The core insight in our setting is the strong heterogeneity of token predictability in world models. What must be estimated is not merely the amount of motion, but the local error incurred when a skipped token update is approximated by reuse or linear prediction. Lemmas 1-3 show why curvature is better aligned with this objective.
> > >
> > > In the actual algorithm, Eq. (7) is the finite-difference counterpart of this smooth-limit quantity, where $v$ and $a$ are estimated from the last three FULL steps. Therefore, **the role of curvature is not post hoc heuristic tuning, but a local approximation to the quantity that governs first-order cache error**. This interpretation is also consistent with the empirical evidence in Table 5, where curvature-based grouping outperforms uniform and random grouping at nearly identical latency.
> > >
> > > We will revise the paper to make this claim explicit and modest: **curvature is justified as a principled local proxy for first-order cache difficulty and it is better aligned than raw difference with the specific notion of cache difficulty that matters for heterogeneous world-model dynamics**.

---

### Official Review · Reviewer_TE6u · 2026-03-13

**Soundness:** 2
**Presentation:** 3
**Significance:** 3
**Originality:** 2
**Overall Recommendation:** 3
**Confidence:** 2

**Summary:**

This paper proposes WorldCache, a training-free acceleration framework tailored for diffusion-based world models. The authors observe that existing caching methods designed for single-modality image or video diffusion models transfer poorly to world models, primarily due to stronger token heterogeneity and non-stationary temporal dynamics. To address this, the paper introduces two core designs: (1) Curvature-guided Heterogeneous Token Prediction (CHTP), which groups tokens using a curvature-based metric and applies reuse, linear extrapolation, or damped updates accordingly. (2) Chaotic-prioritized Adaptive Skipping (CAS), which uses an accumulated curvature-normalized, dimensionless drift signal over chaotic tokens to determine when to execute a FULL backbone computation. Experiments on two diffusion world models (HunyuanVoyager-13B and Aether-5B) demonstrate that the method achieves up to a 3.7× inference speedup while preserving high generation quality and 3D reconstruction capabilities.

**Compliance With Llm Reviewing Policy:**

Affirmed.

**Key Questions For Authors:**

See Strengths and Weaknesses

**Limitations:**

No, the authors can provide additional results on a wider variety of datasets to validate the generalizability of the approach.

**Strengths And Weaknesses:**

**Strengths**

1. The motivation is sound. The authors successfully identify two challenges specific to world models: first, multi-modality and spatial variance result in a long-tailed difficulty distribution for token evolution; second, the temporal dynamics feature non-stationary intervals where a small number of bottleneck tokens dominate error accumulation.

2. The framework logically separates the problem. CHTP tackles the token-level prediction strategy (how different tokens should be cached), while CAS addresses the adaptive skipping policy at the denoising-step level.

3. The empirical results in Tables 1 and 2 indicate that WorldCache is not only faster than the baseline but also outperforms other acceleration methods across multiple quality metrics.

**Weaknesses**

1. The technical novelty is somewhat marginal, lying mostly in an application-specific compositional design. The core components feature caching, token-adaptive processing, and drift-triggered FULL computation overlap substantially with standard approaches.

2. The method relies fundamentally on curvature to measure token predictability, using it as the basis for token grouping and drift normalization. However, the theoretical justification for this choice is limited. Theorem 4.1 primarily establishes a form of scale invariance rather than proving that curvature is actually the most appropriate or optimal proxy for predictability.

3. While the method is evaluated on two recent world models, the overall experimental scope remains narrow. The evaluations are restricted to HunyuanVoyager and Aether under specific resolutions and fixed denoising step configurations. Expanding the experimental setup would better substantiate the paper's core claims.

---

> ### Author Rebuttal · Authors · 2026-03-30
>
> We sincerely appreciate your careful review and valuable insights. Your questions help us clarify critical ambiguities in the ablation study and strengthen the rigor of our work. Below are detailed responses to each concern:
>
> >**W1: About technical novelty.**
>
> For our core contribution / novelty, please see our rebuttal to **Reviewer NsUS `W1 & W2: About core contribution / novelty and properties specific to world models`**.
>
>
> > **W2: Theoretical justification for curvature choice.**
>
>
> We thank the reviewer for this important comment. In the revision, we will clarify the role of curvature by two distinct claims: one for **prediction difficulty** and one for **adaptive triggering**.
>
> **1. Theoretical justification for curvature choice.**
> Role: our use of curvature is first motivated by its physical meaning as the local turning rate of a token trajectory, and we ultimately use it as a **principled local hardness proxy** for cache error. When a denoising step is skipped, the natural low-cost surrogate is the **first-order local linearization** of the token trajectory at the latest FULL step, i.e., $y_i(t)+\Delta y_i'(t)$. The relevant question is therefore how far the true trajectory $y_i(t+\Delta)$ can deviate from this local linear approximation over the skipped interval. Let $y_i(t)$ denote the feature trajectory of token $i$, and assume $y_i$ is $C^2$ on $[t,t+\Delta]$. By Taylor expansion around $t$, the trajectory at $t+\Delta$ equals its first-order linearization plus a second-order remainder, so the first-order cache error is bounded by
>
> $$
> \left\| y_i(t+\Delta)-\bigl(y_i(t)+\Delta y_i'(t)\bigr) \right\|
> \le
> \frac{\Delta^2}{2}\sup_{\xi\in[t,t+\Delta]}\|y_i''(\xi)\|.
> $$
>
> This shows that local cache error is governed by the **second-order departure from local linearity**. Consider the smooth-limit analogue of the curvature score definition,
>
> $$
> \kappa_i(\xi)=\frac{\|y_i''(\xi)\|}{\|y_i'(\xi)\|_2^2}.
> $$
>
> Under a local speed bound $\|y_i'(\xi)\|\le v_{\max}$, we obtain $\|y_i''(\xi)\|\le v_{\max}^2\kappa_i(\xi)$, and therefore
>
> $$
> \left\| y_i(t+\Delta)-\bigl(y_i(t)+\Delta y_i'(t)\bigr) \right\|
> \le
> \frac{\Delta^2}{2}v_{\max}^2\sup_{\xi\in[t,t+\Delta]}\kappa_i(\xi).
> $$
>
> Thus, curvature directly upper-bounds the local first-order cache error under a local speed bound. In the actual algorithm, Eq. (7) is the **finite-difference counterpart** of this smooth-limit quantity, where $v$ and $a$ approximate first- and second-order local dynamics from the last three FULL outputs. Therefore, the theory motivates curvature as a hardness proxy, and Table 5 shows that this proxy indeed separates token difficulty better than uniform or random grouping: CHTP achieves the best fidelity at nearly identical latency. **This supports that curvature-based grouping is identifying tokens with different cache difficulty rather than merely mixing operators.**
>
> Method|PSNR$\uparrow$|SSIM$\uparrow$|LPIPS$\downarrow$|Latency (s)$\downarrow$
> -|-|-|-|-
> Reuse|22.74|0.714|0.336|**86.32**
> Linear|18.01|0.537|0.396|87.07
> Damped|23.76|0.665|0.276|87.51
> Random Group.|22.59|0.710|0.314|86.98
> **CHTP**|**25.76**|**0.791**|**0.227**|86.94
>
> **2. Theorem 4.1 for triggering.**
> Role: once curvature is used to identify hard tokens, dynamic skipping still needs a triggering threshold that is **comparable across timesteps and modalities**. Theorem 4.1 provides exactly this bridge from curvature to a dynamic threshold in CAS. Under global rescaling,
> $$
> y \mapsto sy
> \quad\Longrightarrow\quad
> v \mapsto sv,\;\; a \mapsto sa,\;\; \kappa \mapsto \kappa/s,\;\; \|\Delta y\| \mapsto s\|\Delta y\|,
> $$
> and therefore
> $$
> \kappa' \|\Delta y'\| = (\kappa/s)(s\|\Delta y\|) = \kappa \|\Delta y\|.
> $$
> Hence, $\kappa_i\|\Delta y_{t,i}\|$ is scale-comparable across timesteps and modalities, which is exactly what allows CAS to use a unified trigger threshold. Evidence: Table 6 validates this role directly: Difference-/Norm-Guided triggers improve over fixed intervals but remain inferior because raw non-normalized thresholds are unstable across scales, while Curvature Guided alone also underperforms because hardness without actual displacement is insufficient. CAS performs best because it combines both curvature and actual drift in a scale-comparable metric. **Therefore, Theorem 4.1 justifies the normalized triggering metric used by CAS, not the global optimality of curvature itself.**
>
> Method|PSNR$\uparrow$|SSIM$\uparrow$|LPIPS$\downarrow$
> -|-|-|-
> Fixed Interval|26.18|0.830|0.216
> Difference Guided|26.79|0.824|0.207
> Norm Guided|26.02|0.809|0.217
> Curvature Guided|25.87|0.788|0.236
> **CAS**|**27.10**|**0.881**|**0.198**
>
>
> >**W3 and Limitation: Evaluation on broader settings.**
>
> We conduct more evaluations, including additional world model LingBot-14B, multi-resolution, more denoising steps and longer rollouts. Please see our rebuttal to **Reviewer iAXy `W2 and Q1: Evaluation on more settings.`**.

---

> > ### Author Rebuttal · Reviewer_TE6u · 2026-04-04
> >
> > Thank you for the detailed rebuttal. The added clarifications and additional experiments are helpful for understanding the paper. Nevertheless, I am still inclined to maintain my original score for the following reasons:
> >
> > W1: the rebuttal explicitly states that caching, token-adaptive processing, and drift-triggered recomputation are not individually new, and that the core contribution is instead the identification of a world-model-specific failure regime together with a mechanism designed around it. This clarification is useful, but it also reinforces my original concern to some extent: the method still appears closer to a world-model-oriented reorganization and combination of existing caching ideas. In addition, the evidence for the claimed “world-model-specific” property remains limited. For example, the curvature-variance comparison is based on only a very small number of models, which is not yet sufficient to establish this as a general failure mode of world models.
> >
> > W3: while the new experiments are valuable, they are still concentrated on a relatively limited set of models and metrics, and in several of the added settings the comparison is only against a small subset of baselines.

---

> > > ### Author Response · Authors · 2026-04-05
> > >
> > > We thank the reviewer and clarify both points more explicitly below.
> > >
> > > **1) Our contribution is not merely a reorganization of prior caching ideas.**
> > > Our claim is that prior diffusion-caching methods do not address the harder regime of multimodal world models, where token dynamics are highly heterogeneous and directly applying single-modal policies is unreliable. WorldCache contributes three novel solutions:
> > >
> > > (i) Prior methods use one shared cache/update rule: conservative rules waste compute on easy tokens, while aggressive ones drift on hard tokens. WorldCache is, to our knowledge, the **first** to partition world-model tokens by curvature and assign **heterogeneity-aware predictors (Reuse/Linear/Damped)**. This enables aggressive skipping on easy tokens without sacrificing fidelity on difficult ones.
> > >
> > > (ii) Prior triggers aggregate all tokens, so unstable minorities can be masked by stable majorities and recomputation may start too late. Also the bottleneck subset changes over time, which prior work does not address. WorldCache instead uses **chaotic-prioritized monitoring**, triggering recomputation as soon as the hard subset drifts, enabling earlier and more targeted correction before local errors spread globally.
> > >
> > > (iii) Prior triggering rules use raw feature-difference thresholds, whose scales are not comparable across modalities and timesteps. WorldCache is the **first** to use a **curvature-weighted, scale-invariant trigger**, formalized by Theorem 4.1. This yields more stable and selective skipping.
> > >
> > > Therefore, WorldCache is not recombination of known modules, but three individual novel solutions to the unresolved bottleneck in world-model caching.
> > >
> > > **2) The variance comparison is controlled, not arbitrary.**
> > > We chose **HunyuanVideo-13B -> HunyuanVoyager-13B** because HunyuanVoyager is a world-model variant built on the HunyuanVideo base, so the comparison controls the backbone and isolates the shift from **video generation** to **world modeling**.
> > > We also observe the same trend on additional world-model families, which directly suggests that the phenomenon is general:
> > >
> > >
> > > Model|Curvature Variance
> > > -|-
> > > HunyuanVideo-13B (Video)|13.52
> > > **HunyuanVoyager-13B (World)**|37.76
> > > CogVideoX-5B (Video)|16.92
> > > **Aether-5B (World)**|31.14
> > > Wan2.1-14B (Video)|10.63
> > > **LingBot-14B (World)**|27.35
> > >
> > > **3) We now provide broader experimental evidence.**
> > > Across these settings, **WorldCache consistently improves the quality-efficiency trade-off over prior baselines, rather than benefiting from one or two cherry-picked setups.**
> > >
> > > **(a) New model + multi-resolution.**
> > >
> > > |Method|Resolution|PSNR$\uparrow$|SSIM$\uparrow$|LPIPS$\downarrow$|Latency (s)$\downarrow$|
> > > |-|-|-|-|-|-
> > > DuCa|464×832|19.10|0.716|0.182|448 (1.87x)
> > > ToCa|464×832|18.85|0.704|0.198|455.4 (1.84x)
> > > TaylorSeer|464×832|17.05|0.641|0.286|460.1 (1.82x)
> > > HiCach|464×832|17.82|0.678|0.241|451.6 (1.86x)
> > > TeaCache|464×832|19.85|0.742|0.136|402.8 (2.08x)
> > > EasyCache|464×832|20.28|0.751|0.118|388.1 (2.16x)
> > > **WorldCache**|464×832|**23.24**|**0.812**|**0.079**|**348.7 (2.41x)**
> > > DuCa|720×1280|19.71|0.727|0.012|2190.2 (1.22x)
> > > ToCa|720×1280|19.45|0.714|0.013|2390.7 (1.12x)
> > > TaylorSeer|720×1280|17.60|0.651|0.018|3110.3 (0.86x)
> > > HiCache|720×1280|18.39|0.688|0.015|2820.0 (0.95x)
> > > TeaCache|720×1280|20.49|0.753|0.009|1235.4 (2.17x)
> > > EasyCache|720×1280|20.93|0.762|0.007|1189 (2.25x)
> > > **WorldCache**|720×1280|**23.99**|**0.824**|**0.005**|**1066 (2.51x)**
> > >
> > > **(b) Different denoising schedules.**
> > >
> > > Method|Denoise Steps|PSNR$\uparrow$|SSIM$\uparrow$|LPIPS$\downarrow$|Latency (s)$\downarrow$|
> > > -|-|-|-|-|-
> > > DuCa|70|18.16|0.552|0.452|1016.1 (1.41x)
> > > ToCa|70|17.01|0.453|0.524|1301.8 (1.10x)
> > > TaylorSeer|70|19.82|0.659|0.259|1466.1 (0.98x)
> > > HiCache|70|20.06|0.667|0.247|1348.1 (1.07x)
> > > TeaCache|70|18.05|0.617|0.331|376.1 (3.82x)
> > > EasyCache|70|23.56|0.789|0.167|356.3 (4.03x)
> > > **WorldCache**|70|**25.49**|**0.828**|**0.130**|**351.2 (4.09x)**
> > > DuCa|30|16.13|0.501|0.509|607 (1.00x)
> > > ToCa|30|14.98|0.402|0.581|775 (0.78x)
> > > TaylorSeer|30|17.79|0.608|0.316|926.1 (0.66x)
> > > HiCache|30|18.03|0.616|0.304|852.1 (0.71x)
> > > TeaCache|30|15.61|0.556|0.399|246.9 (2.46x)
> > > EasyCache|30|21.12|0.728|0.235|232.7 (2.61x)
> > > **WorldCache**|30|**22.78**|**0.760**|**0.206**|**223.3 (2.72x)**
> > >
> > > **3) Longer rollouts and memory.**
> > >
> > > Method|Frames|PSNR$\uparrow$|SSIM$\uparrow$|LPIPS$\downarrow$|Latency (s)$\downarrow$
> > > -|-|-|-|-|-
> > > ToCa|81|19.54|0.666|0.137|1505.3 (1.03x)
> > > TaylorSeer|81|17.32|0.608|0.212|1598.5 (0.97x)
> > > HiCache|81|18.23|0.643|0.171|1566.2 (0.99x)
> > > TeaCache|81|20.89|0.701|0.085|694.5 (2.23x)
> > > EasyCache|81|21.81|0.705|0.082|677.4 (2.29x)
> > > **WorldCache**|81|**23.37**|**0.780**|**0.001**|**626.5** (**2.47x**)
> > > ToCa|161|18.81|0.709|0.157|4067.0 (1.01x)
> > > TaylorSeer|161|16.23|0.636|0.217|4615.4 (0.89x)
> > > HiCache|161|17.26|0.676|0.182|4416.9 (0.93x)
> > > TeaCache|161|20.47|0.756|0.118|1694.3 (2.42x)
> > > EasyCache|161|21.81|0.777|0.133|1679.3 (2.45x)
> > > **WorldCache**|161|**22.16**|**0.803**|**0.001**|**1626.0** (**2.53x**)

---

### Decision · Program_Chairs · 2026-04-30

**Decision:**

Accept (regular)

**Comment:**

This work considers token adaptive caching for world models, and uses a physics grounded curvature score to estimate the predictability of the tokens. The resulting method achieves a 3.7x speedup while retaining 98% accuracy. The reviewers acknowledged the strength of the results. The main concerns raised were:

a) the methodology of token adaptive caching is standard in prior works, and this work mainly adapts it to world models. The authors maintain that the main contribution is the design of the curvature score which fits world models after understanding the failure modes of the current caching methods for diffusion models.

b) the theoretical justification for curvature score is limited. The authors present bounds showing how curvature explicitly upper bounds predictability.

In light of the reviews and rebuttal, I recommend weak accept.